# Streamflow and weather conditions of seven small coastal watersheds, British Columbia, Canada, 2013-2019

Maartje C. Korver[1,2], Emily Haughton[1], William C. Floyd[3,4], Ian J. W. Giesbrecht[1,5]

[1]Hakai Institute, Tula Foundation, Heriot Bay BC, V0P 1H0, Canada
[2]*Current address*: McGill University, Department of Geography, Montréal QC, H3A 0B9, Canada
[3]Ministry of Forests, Lands and Natural Resource Operations, Nanaimo BC, V9T 6E9, Canada
[4]Vancouver Island University, Nanaimo BC, V9R 5S5, Canada
[5]School of Resource and Environmental Management, Simon Fraser University, Burnaby BC, V5A 1S6, Canada.

*Correspondence to*: Maartje Korver (maartje.korver@mail.mcgill.ca)

**Abstract.** Hydrometeorological observations of small watersheds of the northeast Pacific coastal temperate rainforest (NPCTR) of North America are important to understand land to ocean ecological connections and to provide the scientific basis for regional environmental management decisions. The Hakai Institute operates a densely networked and long-term hydrometeorological monitoring observatory, that fills a spatial data gap in the remote and sparsely gauged outer coast of the NPCTR. Here we present the first five water years (October 2013–October 2019) of hourly streamflow and weather data from seven small ($< 13$ km$^2$), coastal watersheds. Average yearly rainfall was 3267 mm, resulting in 2317 mm of runoff and 0.1087 km$^3$ of freshwater exports from all seven watersheds per year. However, rainfall and runoff were highly variable depending on location and elevation. The seven watersheds have rainfall-dominated (pluvial) streamflow regimes, streamflow responses are rapid and most water exports are driven by high-intensity fall and winter storm events. Measuring rainfall and streamflow in remote and topographically complex rainforest environments is challenging, hence advanced and novel automated measurement methods were used. These methods, specifically for stream flow measurement allowed us to quantify uncertainty and identify key sources of error, which varied by gauging location. Links to the complete dataset, watershed delineations with metrics, and calculation scripts can be found in Sect. 6 and 7.

## 1 Introduction

Climate and hydrology are major drivers of the physical and ecological connections between land and sea. Understanding the timing and quantities of water, sediment, nutrient, and organic matter fluxes to the ocean can provide the scientific basis for conservation and restoration of coastal environments (Fang et al., 2018), which are vulnerable to anthropogenic pressures and a warming climate (Lotze et al., 2006; Lu et al., 2018). The outer coast of the northeast Pacific coastal temperate rainforest (NPCTR) of North America is characterized by thousands of small watersheds (Gonzalez Arriola et al., 2018) with a wet and mild maritime climate, low to moderately sloped rocky terrain, and open, low productivity forests and wetland ecosystems (Banner et al., 2005). Even though total freshwater inflows to the NPCTR coast are dominated by large drainage basins that have the majority of area located inland (Morrison et al., 2012), small coastal watersheds are thought to play an important role in supporting productive marine food webs and abundant salmon runs (Bidlack et al., 2021). However, the current network of gauging stations in the NPCTR is especially sparse among these small watersheds, due to their remote location and



access limitations (Fig. 1). Consequently, estimates of weather and streamflow from that area are generally derived from models (Moore et al., 2012; Morrison et al., 2012; Hill et al., 2009; Wang et al, 2019; PRISM Climate Group). The Kwakshua Watersheds Observatory (KWO), established in 2013, helps fill this observational data gap by providing continuous and high-resolution hydrometeorological data from seven small ($< 13$ km$^2$) watersheds on the

outer coast of British Columbia (Fig. 1). Besides the hydrometeorology of these watersheds, the KWO also monitors soil hydrology, aquatic biogeochemistry, microbial ecology, and nearshore oceanographic conditions (Giesbrecht et al., 2021) and to date, the observatory has supported studies showing that this area of the NPCTR is a hot-spot of soil organic carbon storage (McNicol et al., 2019) and that high riverine fluxes of dissolved organic matter (Oliver et al., 2017) have a significant effect on the estuarine waters of the NPCTR (St. Pierre and Oliver et al., 2020; St.

Pierre et al., 2021).

Projected changes in the NPCTR's climate - increased mean annual temperature, increased mean annual precipitation and less precipitation as snow - are anticipated to result in a cascade of ecosystem level effects (Shanley et al., 2015; Bidlack et al., 2021) and streamflow regime changes (Déry et al., 2009). The KWO is particularly well-suited to monitor hydrometeorological responses to climate change and to potentially serve within

national reference hydrologic networks (Whitfield et al., 2012), as the monitoring program is set up to be long-term and the gauged watersheds are relatively undisturbed. In addition, the data may be useful for regional forest management decisions (e.g., Kranabetter et al., 2013; Banner et al. 2005) and engineering applications. For example, landslides triggered by heavy rain pose risks to local communities and forestry workers, but warning systems suffer from the lack of long-term quality rain gauges (Jakob et al., 2006). Landslide as well as flooding events are expected

to increase as the frequency of atmospheric rivers has increased in the past decades (Sharma and Déry, 2019) - a trend that is projected to continue (Radić et al., 2015).

This article provides a summary of streamflow and weather conditions between 1 October 2013 and 30 September 2019 from the seven watersheds of the KWO that are representative of the outer coast of the NPCTR. Continuous observations of stream discharge, rain, total precipitation, snow depth, air temperature, wind, relative humidity and

solar radiation from seven hydrometric and fourteen meteorological stations are made available and are presented here to improve our understanding of the hydrology of the region. A novel method to automatically measure streamflow at high flows was developed and implemented, as rainfall events in the NPCTR can be heavy and the manual measurement of rapid streamflow responses are near impossible. A method to quantitatively analyze streamflow uncertainty was developed to accompany this method and special attention is given to the uncertainties

associated with measuring streamflow and rainfall in a coastal rainforest environment with complex topography. All data are provided in hourly timesteps and catchment metadata (location, catchment delineations with metrics) are made available (Sect. 6). High data quality is assured through systematic and thorough quality control methods.

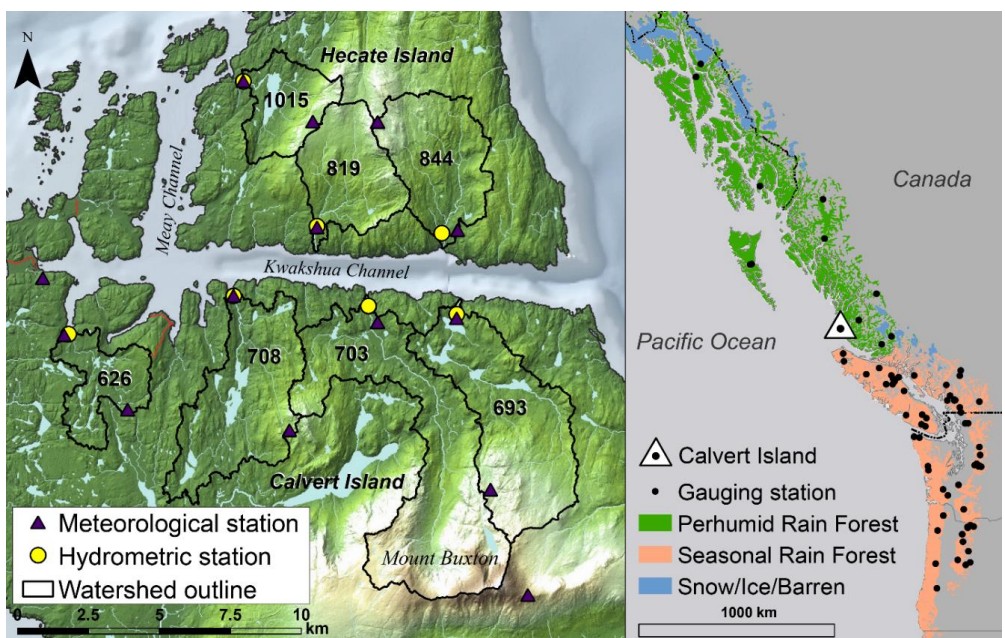

**Figure 1: The seven gauged watersheds at Calvert and Hecate Islands on the outer coast of the northeast Pacific coastal temperate rainforest (NPCTR) of North America. Active government operated gauging stations in small watersheds (Environment Canada stations with gross drainage area < 25 km² and USGS stations with 24 h average summer flow < 1.5 m³ s⁻¹) are shown to indicate the lack of gauging stations measuring outer coast, small watersheds. Terrain relief map was created by Arriola and Holmes (2017). Regional rainforest cover is derived from the original rainforest distribution mapping of Ecotrust (1995), which reflects the rainforest subzones of Alaback (1996).**

**Table 1: Main characteristics of the seven gauged watersheds.**

| Watershed | Area (km²)[a] | Average slope (%)[a] | Mean elevation (m asl)[a] | Max elevation (m asl)[a] | Lakes (count)[b] | Lakes (% area)[a] | Forested (% land area)[c] | Non-forested (% land area)[c] | Non-vegetated (% land area)[c] | Wetlands (% area)[d] |
|---|---|---|---|---|---|---|---|---|---|---|
| **626** | 3.17 | 21.7 | 59 | 160 | 28 | 4.7 | 29.9 | 59.4 | 6.0 | 48.0 |
| **1015** | 3.33 | 34.2 | 132 | 432 | 6 | 9.1 | 79.8 | 8.7 | 2.3 | 23.8 |
| **819** | 4.81 | 30.1 | 248 | 465 | 7 | 0.3 | 75.3 | 21.3 | 3.1 | 50.2 |
| **844** | 5.71 | 32.5 | 218 | 495 | 6 | 0.3 | 84.6 | 12.5 | 2.7 | 35.2 |
| **708** | 7.79 | 28.5 | 93 | 385 | 40 | 7.5 | 66.4 | 21.4 | 4.7 | 46.3 |
| **693** | 9.28 | 30.2 | 230 | 680 | 19 | 4.4 | 65.9 | 26.3 | 3.5 | 42.8 |
| **703** | 12.79 | 40.3 | 325 | 1012 | 53 | 1.9 | 72.0 | 20.3 | 5.7 | 24.3 |
| *Average* | *6.7* | *31.1* | *186* | *518* | *23* | *4.0* | *67.7* | *24.3* | *4.0* | *37.1* |

[a] Reproduced from Gonzalez Arriola et al. (2015).
[b] Lakes are defined as waterbodies > 0.02 ha. From Freshwater Atlas Lakes, British Columbia Ministry of Forests, Lands, Natural Resource Operations and Rural Development, https://catalogue.data.gov.bc.ca/dataset/cb1e3aba-d3fe-4de1-a2d4-b8b6650fb1f6
[c] Forested, non-forested and non-vegetated landcovers were calculated as % of land in each watershed (total watershed area minus total lake area), using the ecosystem classification maps of Thompson et al. (2016).
[d] Reproduced from Oliver et al. (2017), who estimated wetland cover using the Province of British Columbia Terrestrial Ecosystem Mapping (TEM) (Green, 2014; Gonzalez Arriola et al., 2015).



## 2 Site description

Calvert and Hecate Islands are located on the central coast of British Columbia, about 350 km northwest of

Vancouver and 50 km south of Bella Bella. The islands, 325 and 46 km² respectively, are separated by two glacially eroded sea channels, Kwakshua and Meay Channel (Fig. 1), that are connected to Fitz Hugh sound in the east and Hakai Pass in the north. The area's relief ranges from 1012 m a.s.l. in the east of Calvert Island (Mount Buxton) and 495 m a.s.l. on central Hecate Island to relatively low gradient hummocky terrain in the west. The landscape is characterized by extensive wetlands, bog forests, and bog woodlands (Green, 2014; Thompson et al., 2016), with

shallow (typically < 1 m) but organic-rich soils (Oliver et al., 2017), on faulted and folded intrusive igneous bedrock (primarily quartz diorite) (Roddick, 1996). Forest stands are generally short with open canopies and tree composition is dominated by western redcedar, yellow cedar, shore pine, and western hemlock (Thompson et al., 2016; Hoffman et al., 2021). Understory and wetland plants include bryophytes, salal, deer fern, sphagnum mosses and sedges.

Calvert and Hecate Islands are located within the Hakai Lúxvbálís Conservancy (protected area) and the unceded territories of the Haíɫzaqv and Wuikinuxv Nations. The study watersheds have no damns or diversions, no active roads, and little evidence of historic logging except in some shoreline areas which are now well forested. Fire is typically infrequent in the coastal temperate rainforest yet recent work has revealed legacies of long term cultural burning near village sites in this area (Hoffman et al., 2016). More broadly, Indigenous oral history and

archaeological evidence describe at least 13,000 years of human activity in the area (McLaren et al., 2015; McLaren et al., 2018) and active stewardship is ongoing.

The KWO design captures the seven largest (3.0 to 12.8 km²) watersheds draining into Kwakshua and Meay Channel, signified as 626, 708, 703 and 693 (Calvert Island) and 1015, 819, and 844 (Hecate Island). The watersheds are on average 68 % forested, 24 % non-forested but vegetated (including open wetlands), 4 % non-

vegetated (including exposed bedrock), and 4 % covered by lakes (Table 1 ). A total of 159 lakes (> 0.02 ha) are unequally distributed across watersheds (range of 6 – 53 lakes per watershed). Maximum watershed elevations range from 160 to 1012 m a.s.l. and slopes are on average 31 % (Table 1).

Catchment characteristics vary among the seven gauged watersheds (Table 1, Fig. 1). Watershed 703 (12.8 km²) and 693 (9.3 km²) drain the northern aspect of Mount Buxton. Watershed 703 encompasses the mountain top (1012 m

a.s.l.), has steep slopes (40 % average), is sparsely vegetated at high elevations (5.7 % of total area), and has many small lakes and ponds (53, most < 1 ha). Watershed 693 reaches an elevation of 680 m a.s.l., is more gently sloped (30 %) and is distinguished by a chain of four comparatively large lakes (> 5 ha) near the outlet. Watersheds 819 and 844 drain the southern aspect of Hecate Island, are similar in size (4.81 and 5.71 km² respectively) and elevation (max of 465 and 495 m a.s.l. respectively) and exhibit similar landscape features like high vegetation cover (97% of

watershed area) and very few, small lakes (6-7 lakes < 1.5 ha). The watersheds are only different with respect to landcover types, whereas 844 is more heavily forested (85 % vs. 75 %) and 819 has higher wetland cover (50.2 vs. 35.2 %). Watersheds 626 (3.17 km²) and 708 (7.79 km²), located in the lower relief terrain of western Calvert Island (average elevation of 59 and 93 m a.s.l. respectively and < 30 % average slope), have higher lake cover (4.7 and



7.5% respectively). Watershed 626 is typified by multiple small lakes and a relatively large area of exposed bedrock and non-forested vegetated land (e.g. bogs, 59%). Watershed 708 is typified by one centrally located large (30 ha) lake. Watershed 1015 (3.33 km$^2$) has a subdued topography in the west, characterized by a large lake (28 ha.) near the outlet, which is flanked by a steeply sloped ridge in the east (max elevation 432 m a.s.l.). The watershed has a relatively high forest cover (80%).

Each watershed has a hydrometric station near the outlet and a meteorological station at low to mid elevation. In addition, three meteorological stations have been installed at higher elevations between watershed boundaries (Fig. 1, Table 2). Photos of meteorological and hydrometric stations are given in Fig. A1 and A2.

## 3 Methods

### 3.1 Instrumentation and data collection

Seven hydrometric and fourteen meteorological stations were installed in a tight network spanning most of the elevation gradient (Giesbrecht et al., 2021; Fig. 1), as the study area's complex topography leads to rapid changes in climatic parameters over short distances (Beniston, 2006). Station details (installation date, location, and elevation) are provided in Table 2, and sensor inventory and specifications can be found in Table A1. Essential maintenance, replacement, calibration and field checks of the sensors and station infrastructure occurred twice per year in September and May. The stations were connected to a telemetry network facilitating two-way communication and online data storage to overcome accessibility issues. Data transfer and communication between weather and stream gauging stations is controlled with CR1000 data loggers (Campbell Scientific Ltd.) via 900Mhz UHF radios (RF401 Campbell Scientific Ltd) and custom designed, portable, self-powered repeaters. These repeaters transmit to one of two mountain top communication nodes with 2.4 Ghz and 5 Ghz radios (Ubiquiti airMAX devices), with direct links back to station headquarters. Power is supplied to stations primarily through solar panels, with two stations using a combination of solar and micro-hydro. Data are made available in near real-time through satellite internet. The data are managed through a distributed spatial data infrastructure developed by the Hakai Institute to manage, visualize, and share environmental data.

#### 3.1.1 Meteorological stations

All meteorological stations were equipped with a tipping bucket rain gauge (TB4, Hydrological Services America, Lake Worth USA and TR-4 Texas Electronics at 'East Buxton' station) and an air temperature/relative humidity sensor (H2SC3, Campbell Scientific, Edmonton, Canada) with solar radiation shield. Snow depth was measured at three high elevation stations (449 – 740 m a.s.l.) (SR50A: Sonic Distance Sensor, Campbell Scientific, Edmonton, Canada) and a custom-made total precipitation gauge (400 mm diameter pvc pipe, 2000 mm tall, with KPSI 700 pressure transducer) was installed at the 'East Buxton' as well as the 'Reference Station' which has an Alter shield (Campbell Scientific 260-953) to correct for undercatch due to wind. Wind speed and wind direction were measured at stations with topographic exposure (eight of fourteen stations) (05106C-10: Marine Wind Monitor; Campbell Scientific, Edmonton, Canada). Incoming solar radiation was measured at the 'East Buxton' station (SP110 Apogee Limited).

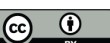

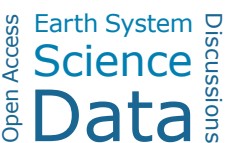

**Table 2:** Station details (installation date, location, and elevation), station specific mean annual runoff and weather parameters (± standard deviation) calculated for 2015/2016 – 2018/2019 water years. Values that are not indicated (-) are not measured at that specific station. Shown are runoff (MAQ), precipitation (MAP), air temperature (MAT), wind speed (MAW), relative humidity (MARH), solar radiation (MAR), and snow depth (MASD). ClimateNA model projections (Wang et al., 2016) of mean annual precipitation with precipitation as snow (PAS) and air temperature (2016 - 2019) are indicated for reference.

| Station | Date installed (m-y) | Lat | Long | Elev. (m a.s.l.) | MAQ (mm) | MAP (mm)[a] | MAP (PAS) - Clim NA (mm) | MAT (°C) | MAT - Clim NA Clim (°C) | MAW (m s⁻¹) | MARH (%) | MASD (m) |
|---|---|---|---|---|---|---|---|---|---|---|---|---|
| RefStn | Nov-15 | 51.6520 | -128.1287 | 42 | - | 2452 ± 335* | 2423 (67) | 8.9 ± 0.3* | 9.4 | § | 82 ± 1* | - |
| SSN626 | Aug-14 | 51.6408 | -128.1219 | 13 | 1750 ± 219 | 2634 ± 322 | 2327 (62) | 8.6 ± 0.4 | 9.4 | - | 83 ± 1 | - |
| WSN626 | Sep-15 | 51.6262 | -128.1018 | 78 | - | 2966 ± 305 | 2525 (76) | 9.2 ± 0.5 | 9.3 | 2.6 ± 0.1 | 80 ± 1 | - |
| SSN693 | Aug-14 | 51.6442 | -127.9978 | 51 | 3066 ± 415 | 3325 ± 414 | 2635 (80) | 8.9 ± 0.4 | 9.4 | 1.1 ± 0.1 | 81 ± 5 | - |
| WSN693_703 | Sep-14 | 51.6106 | -127.9871 | 449 | - | 4392 ± 630 | 3464 (254) | 7.4 ± 0.5 | 7.7 | 3.0 ± 0.2 | 83 ± 1 | 0.6 ± 0.2* |
| SSN703 | Aug-14 | 51.6466 | -128.0257 | 40 | 2804 ± 444 | - | - | - | - | - | - | - |
| WSN703 | Sep-14 | 51.6433 | -128.0228 | 42 | - | 2855 ± 406 | 2464 (70) | 8.2 ± 0.6 | 9.5 | - | 86 ± 1 | - |
| WSN703_708 | Sep-14 | 51.6222 | -128.0507 | 289 | - | 3738 ± 287 | 3148 (156) | 8.6 ± 0.5 | 8.4 | 3.1 ± 0.1 | - | - |
| SSN708 | Sep-13 | 51.6486 | -128.0684 | 12 | 1932 ± 293 | 2440 ± 329 | 2355 (63) | 8.7 ± 0.4 | 9.5 | - | 87 ± 1 | - |
| SSN819 | Aug-14 | 51.6619 | -128.0419 | 79 | 1614 ± 242 | 2515 ± 309 | 2546 (75) | 8.2 ± 0.4 | 9.4 | - | 86 ± 0 | - |
| WSN819_1015 | Sep-14 | 51.6827 | -128.0433 | 331 | - | 2685 ± 335 | 3374 (158) | 7.8 ± 0.5 | 8.6 | 2.8 ± 0.2 | 81 ± 1 | - |
| SSN844 | Sep-14 | 51.6608 | -128.0025 | 35 | 1919 ± 246 | - | - | - | - | - | - | - |
| WSN844 | Jul-15 | 51.6614 | -127.9975 | 90 | - | 2951 ± 362 | 2676 (81) | 8.8 ± 0.4 | 9.4 | - | - | - |
| SSN1015 | Aug-14 | 51.6906 | -128.0653 | 17 | 1502 ± 275 | 2244 ± 310 | 2438 (69) | 8.6 ± 0.4 | 9.4 | - | 87 ± 1 | - |
| East Buxton | Sep-13 | 51.5899 | -127.9752 | 740 | - | 3603 ± 267 | 3865 (461) | 6.0 ± 0.5 | 6.7 | § | 81 ± 2 | 1.5 ± 0.3* |
| Hecate | Jul-14 | 51.6826 | -128.0228 | 477 | - | 2824 ± 282 | 3512 (194) | 7.1 ± 0.5 | 8.6 | 4.2 ± 0.3 | 85 ± 2 | 0.3 ± 0.2* |

[a] Precipitation includes rain+ snow measurements at 'Reference Station' (~5 % precipitation as snow) and 'East Buxton' (~30 % precipitation as snow) and was measured as rain only at all other stations, where snowfall was assumed to be negligible; this was confirmed by the ~5 % PAS at 'Reference Station' and the ClimateNA model estimates of PAS never exceeding 5 % except at 'East Buxton'.

* 2015-2016 ignored due to data gap

** 2018-2019 ignored due to data gap

§ data available but too many data gaps





Air temperature, relative humidity and rain were measured at two meters above ground for all stations, except 'East Buxton' where seasonal snowpacks necessitate the precipitation gauge orifice and air temperature/relative humidity sensor to be installed at 4.5 and 4 meters above ground respectively. Wind speed and direction were measured at 10 m above the ground at 'Reference Station', 8 m at 'East Buxton' and at 5 m at all other stations. Tipping bucket rain gauges (TBRG) were field calibrated semi-annually using a Field Calibration Device (FCD-653, Hydrological Services, Lake Worth, USA) with a 200 mm h$^{-1}$ nozzle rate. For the field test, the FCD was emptied into the TBRG twice and the total number of tips were recorded. The current field specification required the TBRG to be within +/- 6 tips of 202; where 202 tips is the expected number of tips delivered by two applications of the FCD. If the results of the field test exceeded this threshold, a correction factor was applied to the TBRG to adjust the volume per tip ratio. The air temperature/relative humidity sensors were replaced every two years. The sensors were field checked twice per year in between replacements using a 'benchmark' sensor (H2SC3, Campbell Scientific, Edmonton, Canada). In addition, a second temperature sensor (CS 109, Campbell Scientific) was installed at each site as a check and back-up in case of main sensor malfunction. The measurements of the main sensors were corrected if the difference between the benchmark and the main sensor measurements exceeded the sum of their accuracies (± 0.2 °C and ± 1.6 %).

### 3.1.2 Hydrometric stations

Stream water level (stage) was continuously measured at each hydrometric station (starting fall 2013 for watershed 708 and fall 2014 for all other watersheds) and periodic discharge measurements were taken along the range of potential water levels to develop stage-discharge rating curves.

Station locations were selected based on channel stability and potential to measure discharge. Pressure transducers (OTT PLS-L, 2016), anchored to bedrock or boulders with cables armored with steel, were used to measure stage (0-4 m range SDI-12). The mean, maximum, minimum, and standard deviation of five second sampling intervals were recorded every five minutes. Stand-alone water level recorders (Odyssey Capacitance Water Level recorder - Data Flow Systems PTY Ltd 2016, replaced by HOBO Water Level Data Logger - U20L-04 in September 2018) were installed in proximity to each pressure transducer as a backup in case of sensor or data logger malfunction. Relative location of each pressure transducer was regularly surveyed to a benchmark to help identify potential sensor movements. Stream conditions and streambed morphology were monitored through time lapse cameras pointing at each station's stream cross-section and through photos taken during maintenance visits.

Low flows, generally below 0.5 m$^3$ s$^{-1}$, were measured manually using the velocity-area method midsection discharge equation (ISO, 2007), at least once a year during the summer season. Flow velocities, averaged over a 30 s measurement interval, were measured with the Swoffer 2100 propeller type mechanical current meter (Swoffer Instruments Inc., Seattle, USA) or the Sontek Flowtracker acoustic doppler velocimeter (SonTek, San Diego, USA). Manually measuring flows at moderate to high flows was a challenge for multiple reasons: rapid streamflow responses to rain events (generally under 12 hours, Table 4); late fall and winter storm occurrences when field crews were only on site periodically; and safety issues with both accessing the hydrometric stations and taking manual





stream flow measurements at high water levels. Therefore, a novel discharge measurement system was designed based on the salt in solution method as described by Moore (2005): salt in solution (5L water to 1kg salt) is stored in a 1000L IPC tote on site, with a pump to pre-mix the salt solution before a measurement, and another pump to

transfer the solution to a stainless-steel cylinder used to calculate the volume, before it is transferred to a dumping mechanism over the stream that holds up to 40L (Fig. A2). At predefined stages - based on gaps in the stage-discharge rating curve - a signal is sent to release a predetermined volume of salt solution. This volume is targeted to never exceed the most sensitive toxicity threshold of 400 mg L$^{-1}$ (Moore 2004a, 2004b). Upon initiation of the salt solution pump sequence, a second command is sent to a downstream data logger to activate at a minimum two

electrical conductivity sensors (Global Water instrumentation, Inc., College Station, USA; T-HRECS Fathom Scientific Ltd), installed at opposing sides of the stream to capture the passing salt wave. Upon completion of the dump sequence, the EC data are transmitted via the telemetry network for data storage and processing. Recharging of the salt solution reservoir was done manually and a calibration factor (CF), needed to convert EC sensor readings to the relative salinity of the salt solution, was determined prior to refill as well as after refill, with a goal of two CFs

per salt fill. To increase accuracy, a triplicate reading was taken for each CF and each salt in solution measurement was matched with one set of triplicate CF values based on their sampling date. Finally, discharge was calculated using the salt volume data and relative salinity (Moore, 2005) for each EC sensor and for each CF. This yielded six discharge values per measurement, which were averaged to calculate final discharge.

### 3.2 Data analysis and quality control

Characterization of data quality was done by two descriptors, which were stored together with each observation: data processing levels and data quality flags. Data processing levels indicate the status of data handling: unpublished, raw data are termed 'level 1', data subjected to quality control are labelled 'level 2', and 'level 3' refers to derived data products (e.g., gap-filled data). The flagging scheme consists of two tiers: the first tier includes generic flags, e.g., 'Accepted Value' (AV), 'Estimated Value' (EV), 'Suspicious Value Caution' (SVC), and

'Suspicious Value Discard' (SVD) (Henshaw & Martin, 2014). The second tier is a use-case-specific comment on data quality (e.g., background events affecting data values or failed individual quality tests). Any changes or corrections applied to the data are stated in the second tier allowing data users to customize data filling specific to their research objectives. In addition to general quality checks, a quantitative uncertainty analysis of the streamflow data was performed, focusing on discharge measurement and rating curve error. The following section provides

details specific to each dataset for quality assurance and quality control.

### 3.2.1 Weather data

Measured weather data includes air temperature (°C), relative humidity (%), rain (mm), total precipitation (mm), snow depth (m), wind speed (m s$^{-1}$), wind direction (degree), and incoming solar radiation (W m$^{-2}$). Quality assurance procedures involved a combination of visual and automated inspection of the data for inconsistencies,

such as sudden large increases in measurements, as well as outliers. Table A2 details the thresholds used to identify outliers, and the rate of change (ROC) used to identify inconsistencies in the data. All data, except for the snow depth data, were gap-filled using linear regression from nearby station data. For instances when air temperature





readings were faulty, relative humidity (RH) was first converted to vapour pressure and was then converted back to RH based on the appropriate temperature from the nearest station showing the greatest coefficient of
determination. All observations were recorded on a 5-minute time step and aggregated to the hourly time step.

Precipitation data required additional quality assurance procedures due to errors introduced by wind. Tipping bucket rain gauge (TBRG) data from 'WSN626', 'WSN693_703', 'WSN703_708', 'WSN819_1015', 'Hecate', 'East Buxton', 'SSN693', 'Reference Station', and total precipitation data from 'East Buxton' and 'Reference Station' were corrected for undercatch (Yang et al., 2008) using the wind speed adjustment from Legates et al. (2004). In
addition to these corrections, suspect spikes in rain intensity caused by wind-induced tips were identified, initially through visual inspection, but starting from February 2016 through an automated method: when three or more tips occured within a 5 s scan interval, the data were flagged assuming that 3 or more tips within 5 s would yield an extremely unlikely rainfall rate for the area ($> 200$ mm h$^{-1}$). All suspect tips were removed and gap-filled. Obvious spikes in the snow data ($> 1$ m) that did not correspond to increases in total precipitation were removed, but
recurrent gaps in the snow depth data that were caused by sensor failures were too large to be gap-filled.

### 3.2.2 Streamflow data

Stage-discharge rating curves were calculated for each watershed (Fig. 2). Rating curves were updated generally once per year, with each site having between 5 to 50 new measurements.

Stage data were quality controlled and flagged by visual inspection; any spikes or drops and changes in stage not
associated with a rain event were further inspected and where necessary flagged or corrected. Time-lapse photos were also used to assess the turbulence of flows affecting sensor measurements. Data gaps < 30 min were filled using linear interpolation while longer gaps were filled using linear regression of the data from the back-up stand-alone water level sensors that were installed in proximity of the main stage sensors. In the one exceptional case where both water level sensors were malfunctioning (Watershed 819, 1 Dec 2015-17 May 2016), the stage records
of watershed 844, the neighboring watershed sharing many catchment characteristics, were used to estimate stage by linear regression.

Discharge measurement uncertainties were calculated using the Interpolated Variance Estimator (Cohn et al., 2013) for the Velocity-Area method, which estimates uncertainty from the fluctuations in the flow velocity and depth profile, as well as from calibration errors in the width, depth and velocity measurements. Uncertainty estimations for
the automated 'salt solution' discharge measurements were recorded in both a quantitative and a qualitative manner. First, a quantitative % error was calculated by error propagation of the uncertainty in the volume of salt solution, the EC sensor resolution, and the salt solution-electrical conductivity calibration factor (Korver et al., 2018). Measurements with uncertainties higher than 20 % were excluded from further analysis. Second, the EC data were assessed for noise and spikes, caused by turbulence at high flows; where possible spikes were removed, and noisy
data were smoothed. Measurements that contained EC readings so noisy that they could not be corrected, were removed. As an example, peak flows at watershed 703 were not successfully measured due to extremely turbulent flow conditions and hence extremely noisy EC data. Last, an assessment of salt mixing conditions at the downstream





EC sensor site was performed by comparing calculated discharge from the two EC sensors placed within opposing
sides of the river. If calculated discharge differed more than 1 %, mixing conditions were considered poor and,
unless the measurement filled an essential gap on the stage-discharge rating curve, it was excluded from further
analysis.

Rating curves for each station were plotted using locally estimated scatterplot smoothing (LOESS) regression of the
stage-discharge measurements (Fig. 2). The span widths of the LOESS fits were visually assessed and selected.
Discharge data prior to and after high-flow events were analyzed for possible shifts in the rating curves; in case of a
shift, time-lapse videos and photos taken during maintenance surveys were investigated to confirm a concurrent
change in streambed morphology. The curves were extrapolated to minimum and maximum stage of the five-year
stage time-series, estimating minimum stage to equal zero flow, and estimating maximum flow by extrapolating a
power-law equation fitted on the upper section of the rating curves.

Following the methodology proposed by Coxon et al. (2015), rating curve uncertainties were quantified by plotting
95 % confidence intervals (CI) around the curves. CIs were derived from 500 curve fitting results of LOESS
regressions on 500 randomized sets of stage-discharge measurements and their maximum and minimum absolute
measurement errors. Measurement error of a single stage-discharge set was calculated by error propagation of the
discharge measurement uncertainty (described above) and the stage measurement uncertainty. The latter were
calculated from the standard deviation of stage values recorded during the discharge measurement, so that
measurements taken during rapidly falling or rising water levels get assigned higher uncertainties. For each stage-
discharge measurement set on the rating curve, 500 discharge values and 500 standard deviations are predicted and
combined in a Gaussian mixed model, to derive minimum and maximum absolute discharge (95 % CI). Any
minimum flow extending below zero was set to zero. Calculations were done in R and the code has been made
publicly available (Sect. 7). The output is a stage-discharge lookup table with values of discharge, minimum
discharge (95 % CI) and maximum discharge (95% CI) for each mm of stage.

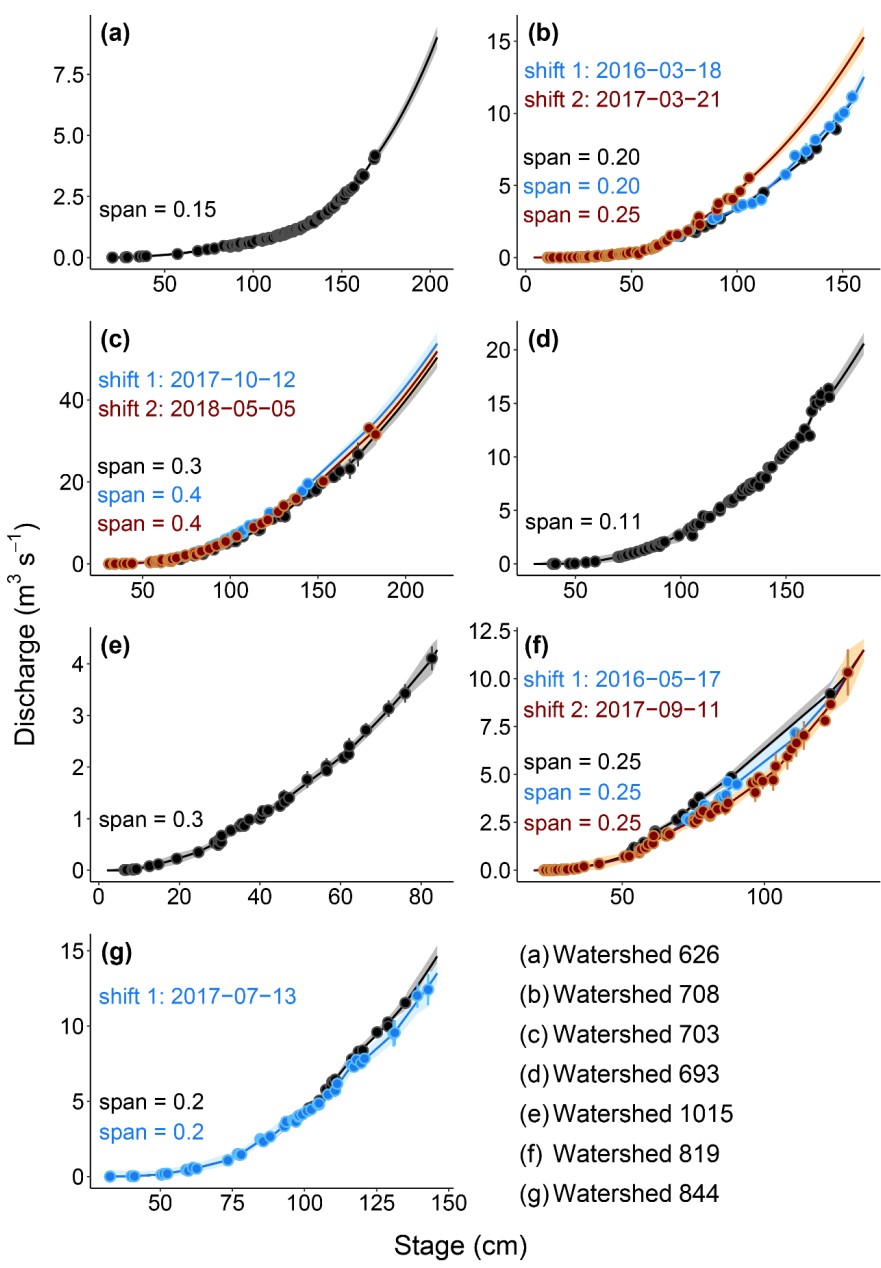

Figure 2: Stage-discharge rating curves with 95 % confidence bands for the seven gauged watersheds. Large storm events caused rating curves to shift at some of the watersheds; dates are indicated for each individual shift. Rating curves were plotted using LOESS regression and span widths ('span') were visually selected for each individual curve.



## 4 Data summary and overview

### 4.1 Weather and climate

Precipitation from 2015 to 2019 averaged 3246 mm over all watersheds, however this may have been slightly underestimated as precipitation was measured as rain only, disregarding snow, at all but two locations (Table 2). Temperatures were generally mild with mean annual temperatures of 8.2 °C ranging from 2.6 °C in December to 14.5 °C in August. Winter was characterized by high windstorms (>10 m s$^{-1}$) from the southeast shifting to calmer winds from the south- and northwest in summer. Snowpacks were recorded at high elevations between December and May all years: persistently above 700 m a.s.l. and intermittently between 400 and 500 m a.s.l. However, there was a high amount of variation in weather conditions by season, year, location and elevation, which will be described in more detail below.

### 4.1.1 Spatial variations

In general, mean annual precipitation increased in a west to east direction, and by elevation, however the rates of change varied due to local conditions. Calvert Island received more precipitation (3353 mm per year) than Hecate Island (2676 mm per year), and for low elevation stations (< 100 m a.s.l.), precipitation increased along a west to east gradient by about 120 mm km$^{-1}$ on Calvert Island and 150 mm km$^{-1}$ on Hecate Island (Fig. 3). Precipitation increased with elevation: only minimally on Hecate Island (about 70 mm per 100 meter) and more pronounced on Calvert Island (about 190 mm per 100 meter) (Fig. 4), likely because the topography of Hecate Island is subdued (max elevation of 495 m a.s.l.) compared to Calvert Island's Mount Buxton (max elevation of 1012 m a.s.l.). There was almost a 50 percent increase in mean total precipitation (3603 vs. 2452 mm) between 'Reference Station' on western Calvert Island (42 m a.s.l.) and 'East Buxton' in the east (740m a.s.l), with 5 % versus 30 % falling as snow at the respective stations. The seasonal snowpack at 'East Buxton' station reached an average maximum depth of 1.4 meters, whereas intermittent snow cover records at lower elevation ('WSN693_703', 449 m a.s.l.) and at the south facing slope of Hecate Island ('Hecate', 477 m a.s.l.) never exceeded 0.6 and 0.3 meters respectively.

Air temperature decreased with increasing elevation at a rate of approximately 0.35 °C per 100 meters (-0.36 and -0.32 °C per 100 m on Calvert and Hecate Islands respectively), from 8.7 °C at sea level to 6.0 °C at 740 m a.s.l. (Fig. 4). This lapse rate varied seasonally with small differences in temperature between low and high elevation stations during summer months (-0.1 °C per 100 m in August) and large differences in winter (-0.5 °C per 100 m in February). All stations recorded average relative humidities between 80 and 87 %, with high elevation stations reaching lower RH's (e.g. minimum of 16 % daily RH at 'East Buxton') than low elevation stations (e.g. minimum of 37 % daily RH at 'SSN708'). General wind directions were uniform across Calvert and Hecate Islands, except for the areas around Mount Buxton which showed local variability. Highest wind speed was recorded at 'Hecate' station (27.3 m s$^{-1}$), which also recorded highest winds on average (4.4 m s$^{-1}$) and station 'SSN693' was, despite being adjacent to a lake, most sheltered (average of 1.1 m s$^{-1}$). Station specific annual weather parameters are summarized in Table 2.


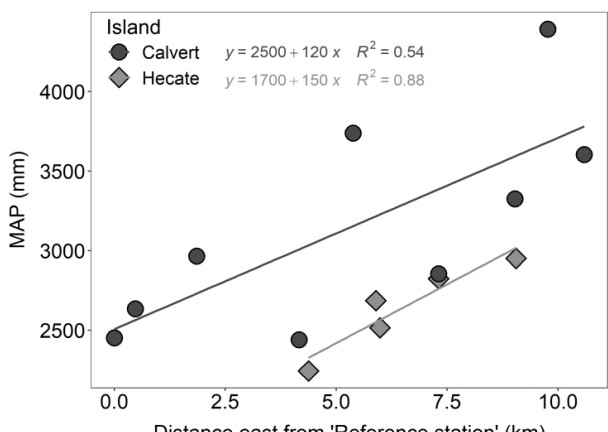

**Figure 3: Mean annual precipitation (MAP) for meteorological stations on Calvert and Hecate Islands, along a longitudinal gradient (expressed as distance east from the westernmost 'Reference Station'). Regression slopes indicate a ~120 mm and a ~150 mm increase in precipitation per km east for Calvert and Hecate Islands respectively. Note that precipitation includes measurements of snow for 'Reference Station' (0 km, 2452 mm) and 'East Buxton' (10.6 km, 3603 mm) stations only, thus values of all other stations might be underestimated.**

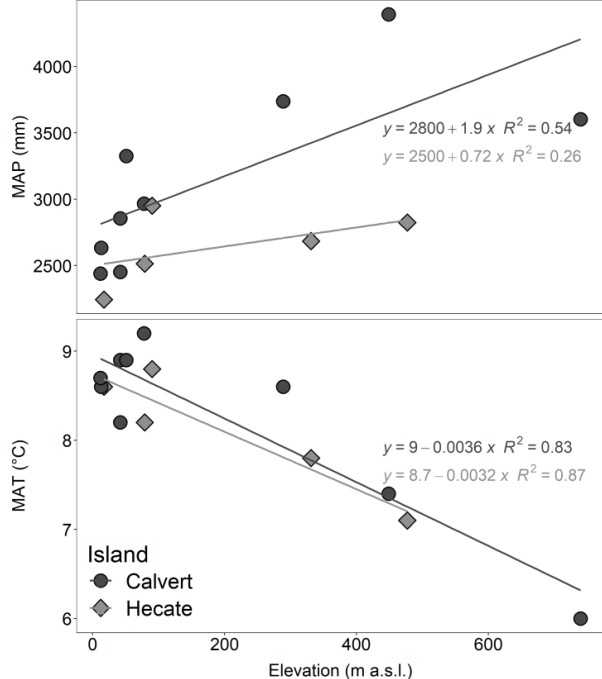


**Figure 4: Mean annual air temperature (MAT) and mean annual precipitation (MAP) by elevation above sea level (a.s.l.), for meteorological stations on Calvert and Hecate Islands. Precipitation increases by approximately 190 and 72 mm and air temperature decreases by 0.36 and 0.32 °C per 100 meters on Calvert and Hecate Islands respectively.**





### 4.1.2 Temporal variations

Calvert and Hecate Islands had wet (Oct – Apr) and comparatively dry (May – Sep) seasons, with the dry season starting abruptly in May but transitioning gradually into the wet season in September (Fig. 5). Although average monthly rainfall was distinctly lower in the dry season (128 versus 320 mm), rain events still regularly occurred: the longest dry periods never exceeded between 9 and 11 consecutive days, usually in July or August. The wet seasons were marked by large inter-monthly as well as inter-annual variations (58 - 524 mm per month), whereas variations

were comparatively small during the dry seasons (22 - 218 mm per month). November was the wettest, and July the driest month of the year (average of 362 and 78 mm respectively). Frequently recurring high-wind storms ($> 10$ m s$^{-1}$) from a predominantly southeastern direction characterized the wet season and low northwest as well as southwest winds ($< 10$ m s$^{-1}$) prevailed during the dry season (Fig. 6). Monthly average air temperature followed an annual cycle ranging between 2.2 °C in February to 14.5 °C in August, with January peaking to 4.3 °C in mid-winter. This

seasonal trend was consistent between years, except for the month of February which varied markedly from year to year (between -0.8 and 7.6 °C). Similar to air temperature, relative humidity followed an annual cycle peaking in August-September (90 %) and reaching a low in February (85%), with relatively more year-to-year variability in winter.

Maximum and minimum air temperatures of 30 °C and -16.3 °C were recorded in September 2016 ('SSN693'

station) and February 2019 ('East Buxton' station). Snowfall was highest in the winter of water year 2017-2018 and lowest in 2015-2016, with maximum snow depths measuring 1.8 and 0.2 meters respectively at 'East Buxton' station. Maximum snow accumulation was usually reached in March (Table 3).

Compared to the 1981-2010 climate normal (8.5 °C, 3602 mm, Table 4), which was calculated for the area of the observatory from a locally downscaled climate model (ClimateNA, Wang et al. 2016), the time period between

October 2014 and October 2019 was slightly cooler and drier (8.1 °C, 3246 mm), except for water year 2016-2017, which was within the short record period anomalously wet and cool (7.6 °C, 3681 mm, Table 3). However, interpreting these deviations from the climate normal warrants caution, as they are more likely explained by discrepancies in the estimation of climate parameters by ClimateNA. When comparing the observatory's data with projected ClimateNA data for the period between 2016 and 2019 (averaged station specific precipitation and air

temperature), they overestimated precipitation at Hecate Island weather stations by 30-700 mm, but underestimated precipitation at Calvert Island by 80-900 mm (Table 2). The precipitation overestimation on Hecate Island is also apparent when comparing watershed averaged modelled climate normals with measured data, which exceed 1000 mm per year (Table 4).





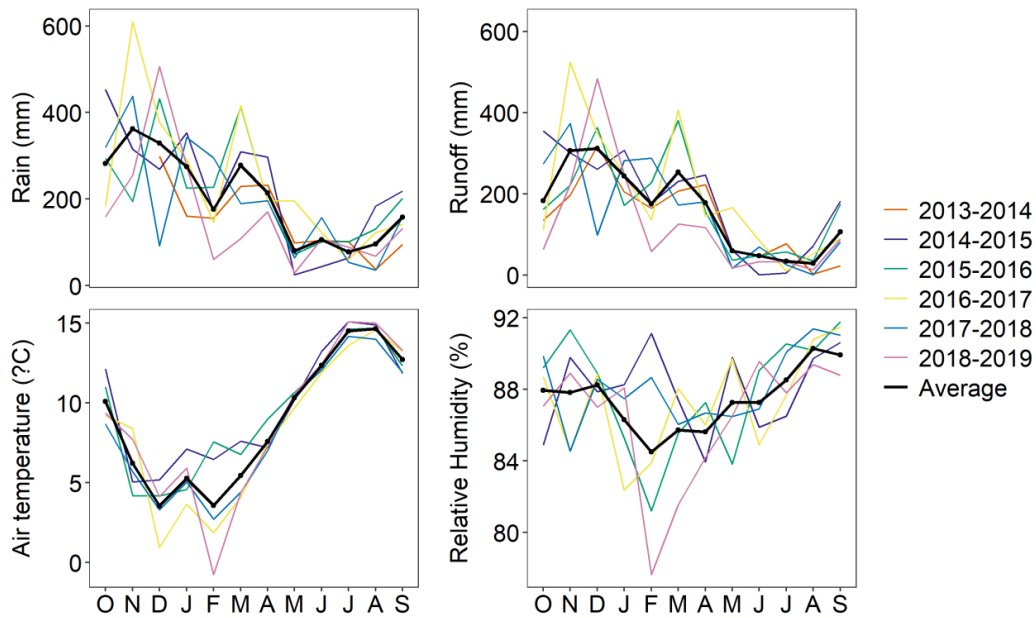

**Figure 5: Seasonal rain, runoff, air temperature and relative humidity by water year. Monthly totals (rain, runoff) and monthly averages (air temperature, relative humidity) from station 'SSN708' are shown, the station with the longest record (installed September 2013, 12 m a.s.l.).**

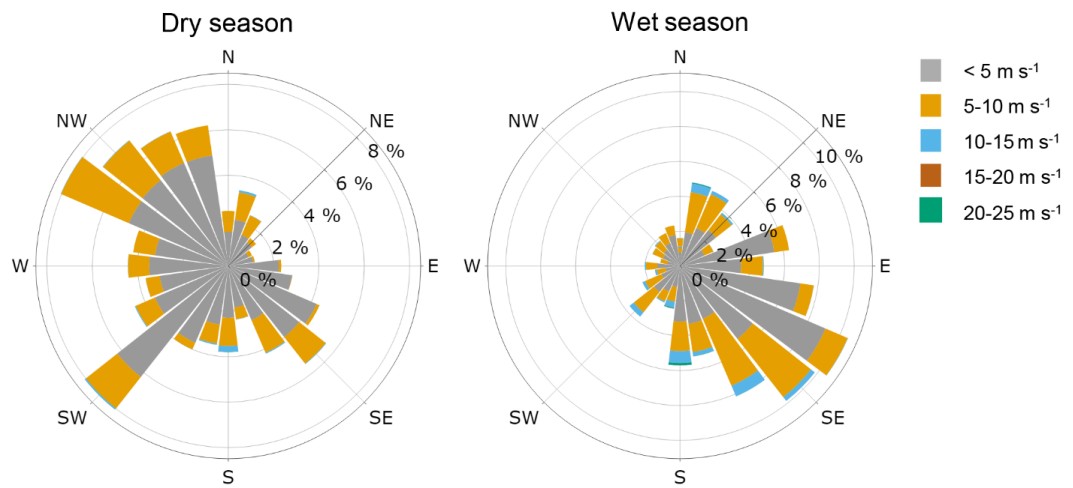

**Figure 6: Average hourly wind speeds and wind directions during the dry (May-Sep) and the wet (Oct-Apr) meteorological seasons. Frequency of occurrence is indicated in % of time for the entire measurement record of 'Hecate' station (Oct 2015 – Oct 2019).**

=


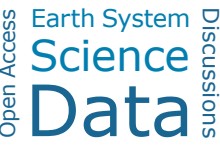


**Table 3: Total precipitation, discharge/runoff, mean discharge, maximum snow depth, mean air temperature, relative humidity (RH), and wind speed by water year (Oct 1st – Sep 30th). Runoff was averaged over all seven watersheds, scaled by watershed area. Discharge (m$^3$) is a total of all seven watersheds and streamflow (m$^3$ s$^{-1}$) was averaged over all seven watersheds. Rain, air temperature and relative humidity data were averaged over all meteorological stations, except 'Reference Station' (no data 2015-2016), and 2014-2015 was not calculated because stations 'WSN6226' and 'WSN844' had not been installed yet. Maximum snow depth was determined from daily aggregated values of 3 h rolling averages.**

| Water year | Total precip.[a] (mm) | Total runoff (mm) | Total discharge (x10$^6$ m$^3$) | Mean streamflow (m$^3$ s$^{-1}$) | Max. snow depth (m)[b] | Mean air temp (°C) | Mean RH (%) | Mean wind speed[c] (m s$^{-1}$) |
|---|---|---|---|---|---|---|---|---|
| **2014-2015** | NA | 2789 | 130.8 | 0.593 | 0.2 (26 Feb) | NA | NA | NA |
| **2015-2016** | 3253 | 2423 | 113.7 | 0.513 | 1.1 (15 Mar) | 8.8 (-10.1 – 26.2) | 84 (23 – 99) | 2.7 (0 - 22.5) |
| **2016-2017** | 3681 | 2724 | 127.8 | 0.579 | 1.8 (17 Mar) | 7.6 (-13.5 – 30.0) | 84 (23 – 100) | 2.8 (0 - 17.2) |
| **2017-2018** | 3279 | 2329 | 109.2 | 0.495 | 1.6 (4 Apr) | 8.0 (-11.0 – 28.0) | 83 (17 – 99) | 2.7 (0 - 22.0) |
| **2018-2019** | 2770 | 1791 | 84.0 | 0.380 | 1.0 (12 Mar) | 8.3 (-16.3 – 26.1) | 82 (17 – 99) | 2.7 (0 - 27.3) |
| *Average[d]* | *3246* | *2317* | *108.7* | *0.492* | *1.4* | *8.2 (-12.2 – 27.5)* | *83 (20 – 99)* | *2.7 (0 - 21.0)* |

[a] Underestimate because all stations, except for 'East Buxton', only measure rain and not rain + snow.
[b] Measured at 'East Buxton' station (740 m a.s.l.).
[c] Measured at 'Hecate' station.
[d] Averaged over water years 2015-2016 to 2018-2019.

## 4.2 Streamflow and catchment processes

Average yearly runoff from all watersheds, scaled by watershed area, was 2317 mm, suggesting that ~30 % of precipitation (3246 mm) is not accounted for in surface runoff. The average flux of freshwater from the seven watersheds was 0.1087 km$^3$ per year. Seasonal runoff patterns followed the wet (Oct – Apr) and dry (May – Sep) patterns of rainfall, with runoff dropping abruptly in May but increasing gradually into the wet season in September reaching highest flows in November and December (Fig. 5). Runoff generated in the wet season accounted for 84 %
of total average yearly runoff.

Freshwater fluxes were dominated by large storm events: 34 % of all freshwater delivered to the ocean occurred during very high flows that were only exceeded for 5 % of the record (≥ P5 flows), and although none of the rivers fully ceased to flow, baseflows were low with P95 flows approaching zero (Table 4). The majority (92 %) of the ≥ P5 flows occurred during the wet season, with 48 % occurring in November and December. In accordance, 98 % of
all days with very low flows (≤ P95) occurred during the dry season of which 52 % occurred during the month of August. Storm events resulted in rapid streamflow responses: average lagtimes (the time from peak rain to the start of rise in streamflow) were less than 8 h for most watersheds; average peaktimes (the start of rise in streamflow to peak discharge) were generally under 12 h (Table 4). However, runoff responses varied across the study area: this is illustrated by Figure 7 which shows the hydro-hyetograph of a large rainstorm which produced 114 mm over 3 days
and reached highest rain intensity (74 mm in 24 h) on 16 October 2017. Discharge peaked at 50 and 20 m$^3$ s$^{-1}$ at watersheds 703 and 693, but never exceeded 12 m$^3$ s$^{-1}$ in the other rivers. This can only be partially explained by differences in watershed size; those watersheds draining large lakes (1015 and 708) or a chain of lakes (693), had relatively subdued peak flows and longer lasting receding limbs. In addition, whereas peak flows generally showed a positive relation to watershed size, high flow values (P5) of watersheds 708 and 1015 were subdued compared to
similar, or smaller sized watersheds (Table 4). Further, lagtimes and peaktimes of watersheds 1015 and 693 (~20

=





and ~25 hours) were much higher than those of the other watersheds, including watershed 708; this can be explained by the relatively small contributing watershed area upstream of 708's large lake, whereas the lakes at 1015 and 693 are located close to the watershed outlets.

Variations in discharge volume and streamflow responses among the seven watersheds can be explained by watershed characteristics but also by the spatial variation in precipitation (Sect. 4.1.1). For example, although freshwater fluxes (total volume) were directly related to watershed size, runoff (area scaled yields) varied greatly between 1502 and 3066 mm per year for watersheds 1015 and 693 respectively, showing a clear spatial pattern of increasing runoff from west to east, with lower overall runoffs on Hecate Island. Discharge patterns mirror those of rain and total precipitation where inputs vary between 2465 and 3774 mm per year at watersheds 1015 and 693

respectively. There are potentially three factors that contributed to these differences; first the presence of Mount Buxton with peak elevations of 1012, 680, and 385 m a.s.l. in watershed 703, 693 and 708 respectively, which resulted in elevated precipitation lapse rates on eastern Calvert Island. Second, the majority of large storm events occurred during the wet season when dominant wind direction was from the southeast, and thus there was potential that Hecate Island and western Calvert Island were being affected by a rain shadow from Mount Buxton. Last, the

steep relief, exposed bedrock, and sparsely vegetated and shallow soils at high-elevation areas of Mount Buxton contributed to rapid runoff generation and elevated runoff coefficients of 77 and 81 % (versus 60 to 66 % for the other watersheds, Table 4). However, presented watershed averaged precipitation estimates do not include any stations above 740 m a.s.l., and thus are potentially underestimating total precipitation, resulting in overestimated runoff coefficients in watershed 703 and to a lesser extent in 693.

A muted freshet from seasonal snowpacks contributed to stream flow in watersheds 703 and 693. Visual inspection of the hydrograph of watershed 703 showed daily fluctuations of streamflow during dry conditions from April to June, with peak flows occurring ~12 hours after maximum daily air temperature (Fig. 8). These streamflow fluctuations were, compared to peak flows after rainfall events, small (increases of a maximum of 0.2 $m^3$ $s^{-1}$, versus up to 1 to 40 $m^3$ $s^{-1}$ during rain events) and could not be distinguished during rainy periods, suggesting snowmelt

makes up a small component of annual runoff generation. However, the Mount Buxton snowpack has the potential to contribute to peak flows during rain-on-snow events, especially in watersheds 703, 693 and to a lesser extent in 708.

Based on the general climatic conditions and hydrographs, these watersheds have rainfall-dominated (pluvial) streamflow regimes with peak flows occurring from the early fall through to mid-winter and low flows in the

summer dry months (Moore et al., 2012). This contrasts to the freshwater fluxes from one of the major drainage basins within the larger study area, the Wannock river catchment (3900 $km^2$, ~50 km east of Calvert Island), which has a snow (nival) and glacial melt dominated regime, with the highest sustained flows in the summer months. Total Wannock river freshwater fluxes reached on average 9.5742 $km^3$ per year which is around a factor 100 higher than the fluxes from the seven Calvert and Hecate Island watersheds combined (0.1087 $km^3$ per year). However, yields

(scaled by watershed size) were comparable (2317 vs. 2455 mm) (Wannock River daily discharge data, 2021).

Despite major differences in flow regimes across the region as described above, Wannock river peak flows generally occurred in the late fall and early winter when atmospheric rivers make landfall (Wannock River annual instantaneous extreme discharge data, 2021), contributing up to 44 % of annual runoff in coastal watersheds (Sharma and Déry, 2019). These atmospheric rivers, combined with warm temperatures and strong winds can melt

early season snow and glaciers at higher elevations. It is during such storms that freshwater discharge was synchronized across the region, with corresponding high material flux to the marine ecosystems. It is expected that current yields and streamflow regime will change due to accelerated glacial loss in the Wannock basin (Menounos et al., 2019), a projected increase in the region's mean annual precipitation with less falling as snow (Shanley et al., 2015), and more frequent and intense atmospheric river events (Déry et al., 2009).

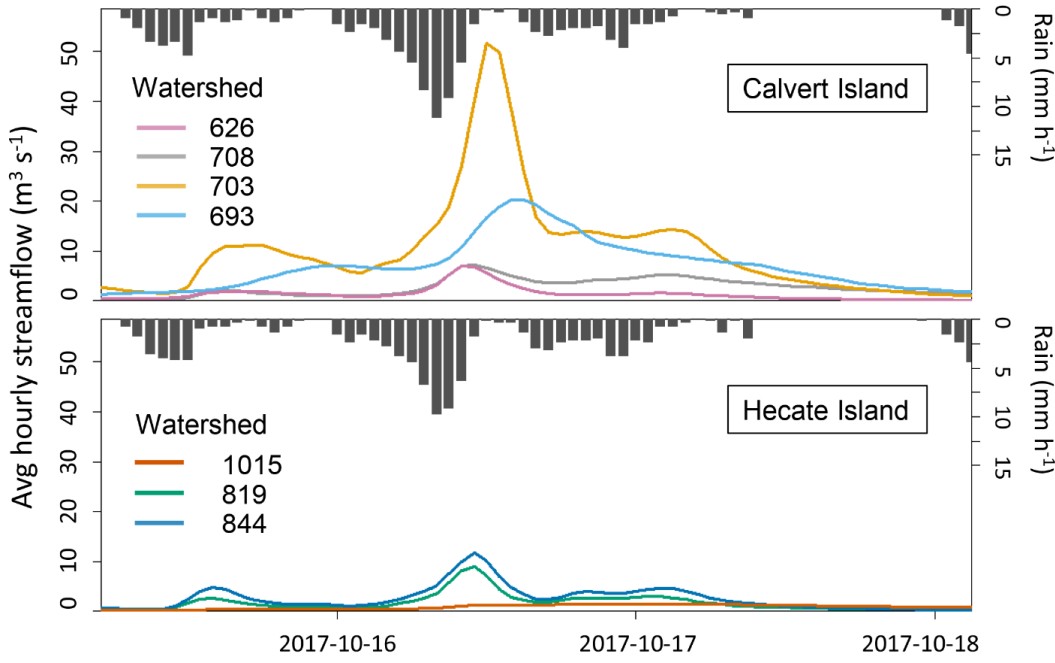

**Figure 7: Hydro-hyetographs for the seven gauged watersheds during a large storm event (114 mm of rain in 3 days). Rain measured at 'SSN708' and 'SSN819'.**



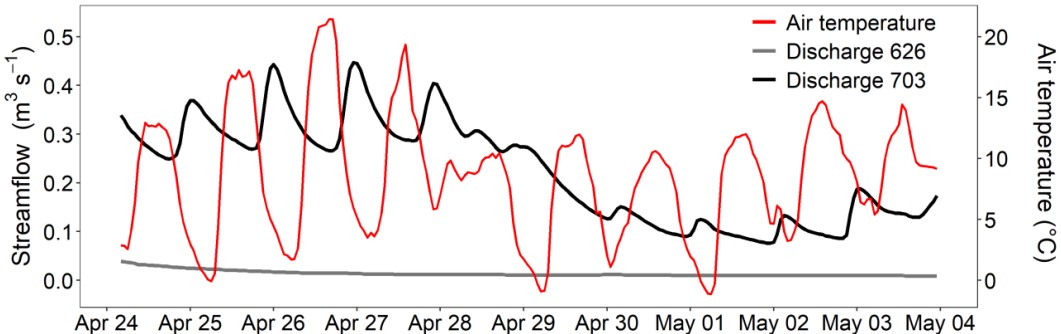

**Figure 8: Discharge and air temperature at watershed 703 during 10 days without rainfall in the snowmelt season (measured at 'SSN703' and 'WSN703' respectively). Daily fluctuations in 703 streamflow indicate snowmelt contributions: peak flows occur ~12 h after daily maximum air temperatures. Discharge at watershed 626 (low elevation watershed with no snowpack), shown for reference: stable recession indicates no rainfall inputs during this period.**






Table 4: Mean annual precipitation (MAP), air temperature (MAT), and streamflow (MAQ) by watershed, averaged over water years with complete precipitation station records (Oct 2015 – Oct 2019), MAQ is indicated by total discharge (m³), total runoff (mm), and average streamflow (m³ s⁻¹), with uncertainty estimations for total discharge and runoff. The runoff coefficient is calculated by MAQ/MAP; flow-duration exceedance probabilities are indicated for low (P95) and high (P5) flows with % of annual discharge originating from >P5 events. Mean lag- and peaktimes (time from peak rain - start of rise in hydrograph and start of rise in hydrograph - peak discharge respectively) were calculated from 50 rain events of varying sizes occurring throughout the record period during both wet and dry seasons, that started from baseflow conditions and had single peakflows in all rivers. Precipitation and air temperature were averaged over the mean annual records of the stations located inside and at the border of each watershed; averages for watersheds 693 and 703 also include measurements from 'East Buxton', which is located just outside watershed boundaries but close enough to be representative of the watersheds' high elevation areas. Watershed precipitation with precipitation as snow (PAS) and air temperature climate normals (1981-2010) were calculated by averaging station specific, modelled ClimateNA data (Wang et al., 2016).

| Watershed | MAP (mm)[a] | MAP - (PAS) 1981-2010 (mm) | MAT (°C) | MAT 1981-2010 (°C) | MAQ (x10⁶ m³) | MAQ (mm) | MAQ uncertainty (%) | MAQ (m³ s⁻¹) | Runoff coef. (-) | P5 (m³ s⁻¹) | P95 (m³ s⁻¹) | Annual Q vol. from >P5 events (%) | Mean lagtime (h) | Mean peaktime (h) |
|---|---|---|---|---|---|---|---|---|---|---|---|---|---|---|
| 626 | 2800 | 2745 (80) | 8.9 | 8.9 | 5.6 | 1750 | -18/+23 | 0.176 | 0.62 | 0.820 | 0.003 | 37 | 7.0 | 12.8 |
| 708 | 3089 | 3229 (115) | 8.2 | 8.6 | 15.1 | 1932 | -19/+23 | 0.477 | 0.63 | 1.641 | 0.002 | 27 | 7.2 | 10.8 |
| 703 | 3647 | 3829 (218) | 7.7 | 8.2 | 35.9 | 2804 | -23/+39 | 1.137 | 0.77 | 5.110 | 0.020 | 34 | 6.3 | 11.8 |
| 693 | 3774 | 3886 (243) | 8.0 | 8.1 | 28.5 | 3066 | -21/+41 | 0.902 | 0.81 | 3.961 | 0.005 | 34 | 21.3 | 27.3 |
| 1015 | 2465 | 3324 (117) | 8.7 | 8.7 | 5.0 | 1502 | -32/+41 | 0.158 | 0.61 | 0.678 | 0.001 | 33 | 17.4 | 21.4 |
| 819 | 2675 | 3724 (168) | 8.2 | 8.5 | 7.8 | 1614 | -33/+74 | 0.246 | 0.60 | 1.156 | 0.001 | 37 | 5.8 | 11.5 |
| 844 | 2888 | 3683 (177) | 7.5 | 8.5 | 11.0 | 1919 | -32/+77 | 0.347 | 0.66 | 1.684 | 0.007 | 37 | 5.4 | 11.2 |
| *Average or Total** | *3246* | *3602 (179)* | *8.1* | *8.5* | *108.9** | *2317* | *-24/+43* | *0.492* | *0.67* | *2.150* | *0.005* | *34* | *10.1* | *15.3* |

ᵃ Annual precipitation is calculated from rain- as well as snowfall measurements at watersheds 693 and 703 (including total precipitation from 'East Buxton'), but from rainfall measurements only at the other lower elevation watersheds where snowfall was assumed to be negligible; this was confirmed by the ~5 % PAS at 'Reference Station' (42 m a.s.l., outside watershed boundaries) and the ClimateNA model estimates of PAS never exceeding 5 % (see Table 2 for station specific PAS model estimates).



## 5 Data uncertainties

### 5.1 Uncertainties in the weather data


Rainfall data that were corrected for wind-induced undercatch were on average 12 % greater than the raw data. The largest corrections were applied at 'WSN693_703' (13 %), which was exposed to some of the highest recorded windspeeds (Table 2). In contrast, station 'SSN693' was at a much lower elevation with lower exposure, resulting in overall corrections of 7 %. Some stations exposed to wind required additional anchoring and thus had increased

uncertainty due to wind induced tips between 2014 and 2018 ('WSN693_703', 'WSN703_708', 'Hecate'). This was corrected using a tip threshold (> 3 tips per 5 s) to remove faulty tips. However, it was difficult to discern whether some of these tips represented real events even when using adjacent stations as comparison. Following proper anchoring in 2018, 1 to 2 % of the annual rainfall record was flagged as suspect and cleaned. Tipping bucket calibrations indicated there were no shifted values over the course of the study period, but site-specific issues with

tipping buckets included blown off lids ('WSN703_708'), blockages from debris ('WSN703' and 'WSN844') and wiring damage by wolves causing false tips ('819_1015'). Resulting issues were addressed and data gaps were filled using linear regression of nearby station records.

The majority of weather station data presented here only measured rainfall (twelve out of fourteen stations, Table 2), and thus there was likely an underestimation of precipitation across the study area. All stations above 400 m a.s.l.

measured snow depth to assess the relative contribution of snow, but these data were not used to correct rainfall at these locations. However, according to ClimateNA modelled data, snow made up less than 5 % of annual precipitation at all stations (except 'East Buxton' at 740 m a.s.l.) including those between 400 and 500 m a.s.l. It is therefore likely that any errors associated with not measuring total precipitation at all sites is less than the variation in precipitation across the study area.

Snow depth data were corrected by removing obvious spikes, induced through false returns during heavy snowfall and wind, but large data gaps due to sensor failures were not filled with this dataset. It should be noted that the SR50 snow depth sensors had a high rate of failure (1 to 2 years), and thus sensors were replaced annually. In contrast, the air temperature, relative humidity and solar radiation data were without issues. The anemometers occasionally froze in winter causing wind direction records to get 'stuck'. However, these issues were usually short-lived as sensors

would melt during daytime.

### 5.2 Uncertainties in the streamflow data

Stream flow measurements are affected by many sources of uncertainty, and thus it is important to both identify and quantify these errors. One potentially major source of error which can be difficult to quantify, is the lack of high-flow discharge measurements necessitating the extrapolation of a rating curve from the highest measured flow up to

the highest measured stream stage (Coxon et al., 2015; Domeneghetti, 2012). However, the automated salt dilution measurement system used here enabled the collection of hundreds of measurements taken along the near-full range of stage values in the rating curve for most of the watersheds, therefore allowing us to quantify discharge uncertainty generally not reported.





Channels were stable in watersheds 626, 693 and 1015 for the duration of the data presented, but large storm events
resulted in rating curve shifts in watersheds 708, 703, 819 and 844. These watersheds all experienced one to two
rating curve shifts within the five-year data record (Fig. 2). However, the automated salt dilution measurement
method allowed regular and ample discharge measurements before and after big storms, and thus shifts could be
tracked with minimal rating curve data gaps. In addition, peak flows were measured at the upper end of the rating
curve in most watersheds. For all watersheds, less than 0.12 % of discharge data points were calculated from the
extrapolated part of the rating curve; in other words, less than 2.5 days of the five-year discharge time series are
estimated and not measured. On a volumetric basis, between 1 and 3 % of total water discharge from watersheds
626, 693, and 819 was calculated from the extrapolated parts of rating curves, and below 1 % for all other
watersheds.

Despite having discharge measurements over the majority of stage measurements in the rating curves, there was
significant error in the discharge dataset presented, with mean annual runoff from the seven watersheds potentially
being overestimated by 24% and underestimated by 43%. Uncertainty varied by watershed (Table 4), with amounts
depending on the spread and measurement uncertainties of individual discharge data points at different positions
along the rating curves (affecting the widths of confidence bands). Underlying causes were related to channel
stability, turbulence, downstream mixing of salt solution, and general uncertainties introduced by equipment
accuracies, which are described in more detail below.

Discharge measurement error was always below 20 % at low flows, and generally under 5 %, but occasionally no
more than 15 % at high flows using the automated salt-dilution method. The largest factor contributing to the
uncertainty of low flow measurements using the velocity-area method, was uncertainty in the velocity readings:
despite careful selection of river cross-section sites, ideal sites with minimal flow velocity variation were sparse due
to the rivers' steep gradients and complex streambeds. The largest factor contributing to salt-dilution measurement
uncertainty was error associated with the salt solution-electrical conductivity calibration factor, which was sensitive
to the salt solution's mixing prior to transfer, temperature, and the precision of the calibration equipment (~70 % of
total measurement error). The next largest factor was error associated with determining the salt solution dump
volume, which due to the system being automated, had to be determined indirectly from pressure measurements
(using a pressure transducer inside a stainless-steel collector) and could therefore vary with salinity and within the
limits of sensor accuracy. This uncertainty increased with decreasing volume (~30 % of total measurement error).
Further on, discharge measurement error was increased by incomplete mixing of salt solution, or excessive noise in
the EC data at downstream sensors. However, most measurements affected were removed from further analysis,
except for measurements at the upper end of the rating curves where no other data were available (e.g., high flows at
watersheds 703 and 1015 reaching up to 15 % measurement uncertainty).

Selecting a stage value to match a discharge measurement for rating curve plotting was difficult when flows were
either turbulent and/or the stream level was rapidly rising or falling during the discharge measurement. This error
was quantified by propagating the standard deviation of stage values recorded during a discharge measurement to
total discharge uncertainty. Fluctuating stage values were mostly a concern for high-flow measurements at





watershed 819, where stage values could be noisy and rapidly increase or decrease within a 5 to 15 minutes measurement interval, increasing discharge measurement uncertainties by on average 2 % (range of 0 to 9 %). This was also a concern at watershed 844, but to a lesser magnitude (uncertainty increases of 0 to 3 %). All other watersheds were only minorly affected (typically < 1 %).

Rating curve confidence intervals are narrow, resulting in low overall discharge uncertainty, if 1) discharge
measurement uncertainties are low, and 2) there is little spread in the rating curve data points. These conditions were met for watersheds 626 and 708, resulting in an overall uncertainty in yearly runoff of -18/-19 % (lower CI) and +23 % (higher CI) (Table 4, Fig. 2). The rating curves of watersheds 703 and 1015 were of high quality at low to mid flows, but confidence intervals widened above 15 and 1 $m^3$ $s^{-1}$ respectively, because of increased measurement uncertainties related to incomplete salt solution mixing at the downstream sites. Measurements taken at watershed
693 all had low uncertainties (< 5 % except at very low flows), but nonetheless a spread in the data (> 3 $m^3$ $s^{-1}$) increased confidence intervals suggesting there were sources of error unaccounted for in the quantitative uncertainty estimations. Measurements taken at mid to high flows at watersheds 819 and 844 (> 4 and 5.5 $m^3$ $s^{-1}$ respectively) had the highest uncertainty due to incomplete salt mixing at the downstream EC sensors and rapidly fluctuating stage during high flows. In addition, unstable channel conditions at watershed 819 caused a rating curve shift once
per year, and in some instances the complete rating curve could not be rebuilt before the next shift. All above issues resulted in an overall uncertainty in yearly runoff of -32/-33 % (lower CI) and +74/77 % (higher CI) for watershed 844 and 819 respectively.

## 6 Data availability

All discharge and weather data presented in this paper are available at https://doi.org/10.21966/J99C-9C14 (Korver
et al., 2021) and watershed delineations with watershed metrics can be found at http://dx.doi.org/10.21966/1.15311 (Gonzales and Giesbrecht, 2015). Both data packages include a readme file detailing data file content and contact information for further details.

## 7 Code availability

Calculations were done in R: the rating curve script is available at https://github.com/HakaiInstitute/RatingCurve
and the scripts used for weather data QC can be found at https://github.com/HakaiInstitute/wx-tools.

## 8 Conclusions

The hydrometeorological data of the Kwakshua Watersheds Observatory fills a critical data gap on the outer coast of the northeast Pacific coastal temperate rainforest (NPCTR) of North America and over time will help to better understand the hydrology of the region. The hydrometeorological dataset presented is part of a greater
interdisciplinary effort to better understand this region, including oceanography, biodiversity, ecology, climate and watershed processes. High quality data are assured using state-of-the-art technology and thorough data quality procedures. Discharge data presented here are unique because the novel method of measuring streamflow accounts for measurement error from various sources, which is often not quantified in publicly available datasets. The dataset



also highlights the difficultly of measuring streamflow in small, turbulent streams and identifies key sources of

uncertainty which should be included when using these data for analysis and modelling.



## 9 Appendix

**Table A1: Sensor specifications and sensor inventory by station.**

| Measurement | Station | Sensor specification |
|---|---|---|
| Air temperature and relative humidity | RefStn, SSN626, WSN626, SSN693, WSN693_703, WSN703, WSN703_708, SSN708, SSN819, WSN819_1015, WSN844, SSN1015, East Buxton, Hecate | H2SC3, Campbell Scientific, Edmonton, Canada |
| Wind speed and direction | RefStn, WSN626, SSN693, WSN693_703, WSN703_708, SSN1015, East Buxton, Hecate | 05106C-10: Marine Wind Monitor, Campbell Scientific, Edmonton, Canada |
| Solar radiation | East Buxton | SP110 Apogee Limited |
| Rain | RefStn, SSN626, WSN626, SSN693, WSN693_703, WSN703, WSN703_708, SSN708, SSN819, WSN819_1015, WSN844, SSN1015, East Buxton, Hecate | TB4, Hydrological Services America, Lake Worth USA and TR-4 Texas Electronics (East Buxton station) |
| Total precipitation | RefStn, East Buxton | Custom made: 400 mm diameter pvc pipe, 2000 mm tall, with KPSI 700 pressure transducer and Alter shield, Campbell Scientific 260-953 |
| Snow depth | East Buxton, WSN793_703, Hecate | SR50A: Sonic Distance Sensor, Campbell Scientific, Edmonton, Canada |
| Water level | SSN626, SSN693, SSN703, SSN708, SSN819, SSN844, SSN1015 | OTT PLS: Pressure Transducer, OTT Hydromet |

**Table A2: Thresholds used for quality control procedures of the weather data: maximum, minimum, rate of change (ROC) and maximum timestep of constant value. Outlier data were flagged or where necessary, corrected and gap filled.**

| Variable | Unit | Maximum | Minimum | ROC limit | Time steps to flag constant value |
|---|---|---|---|---|---|
| Air temperature | °C | 40 | -30 | 10 | 96 |
| Relative humidity | % | 100 | 0 | 30 | 96 |
| Wind speed | m s$^{-1}$ | 35 | 0 | n.a | 96 |
| Wind direction | degrees | 360 | 0 | n.a | 12 |
| Snow depth | m | n,a | 0 | 1 | 96 |
| Solar radiation | W m$^{-2}$ | 1368 | 0 | 1450 | 96 |
| Precipitation | mm | 30 | 0 | n.a | n.a |

**Figure A1: Images of meteorological stations (continued next page). Photos by Shawn Hateley and Bill Floyd.**

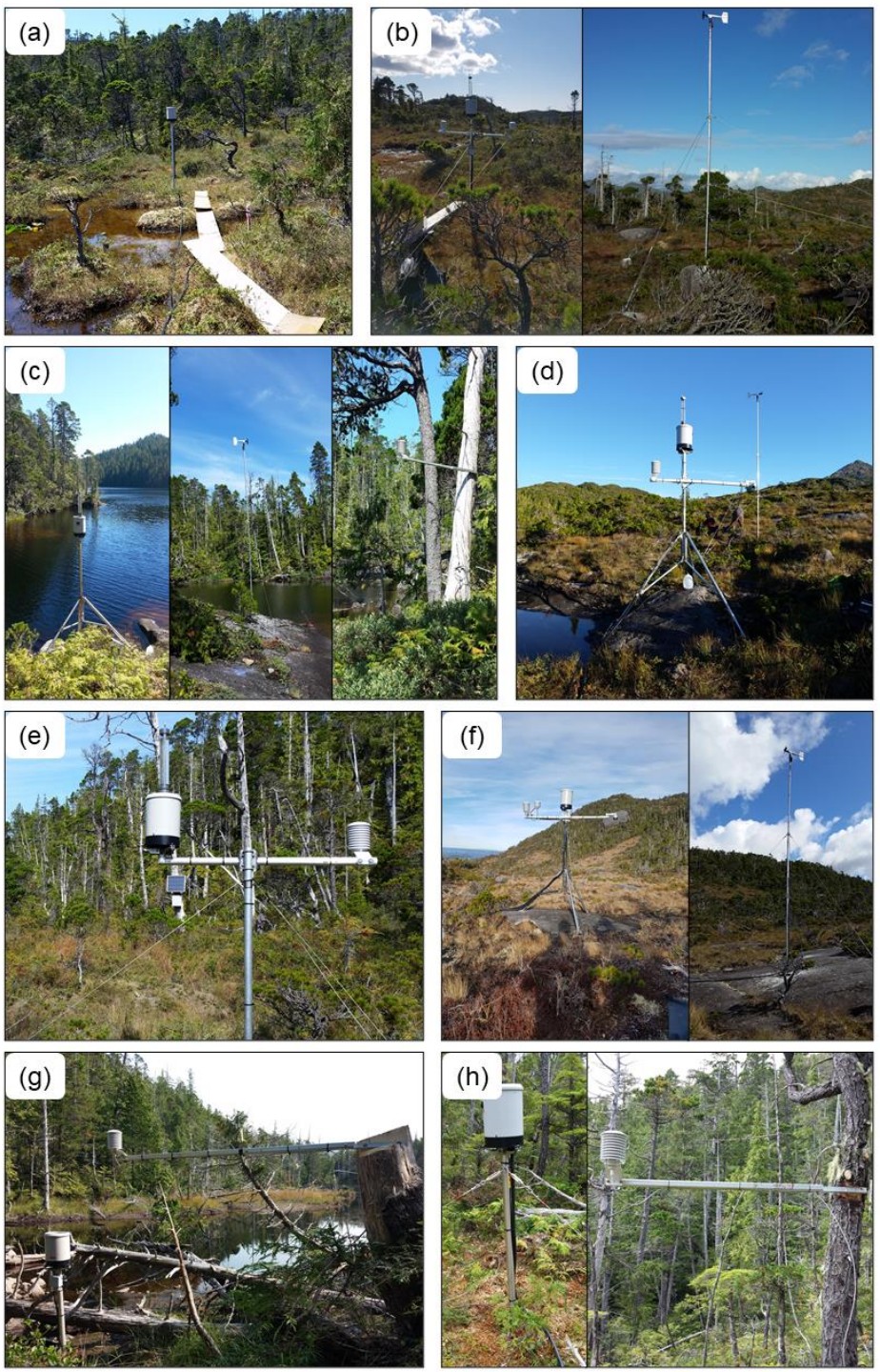

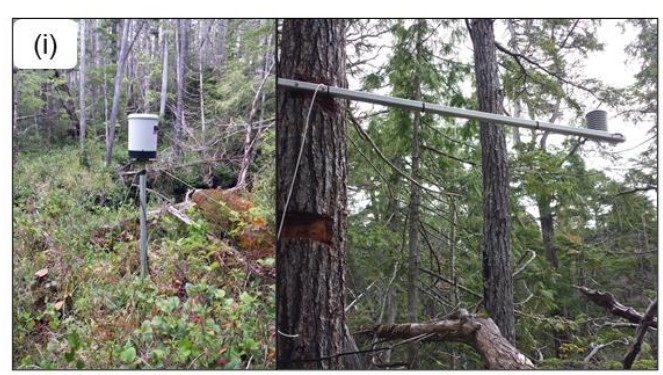

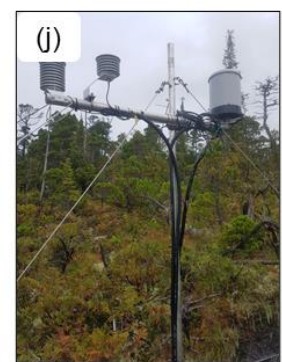

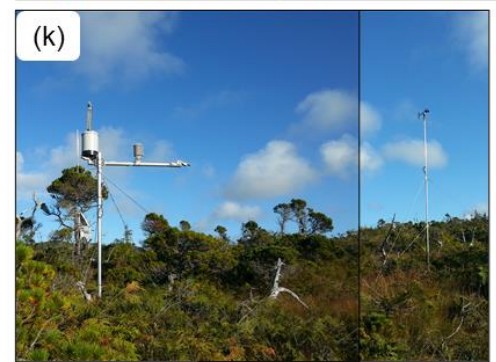

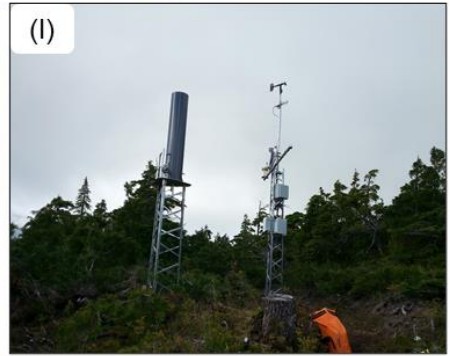

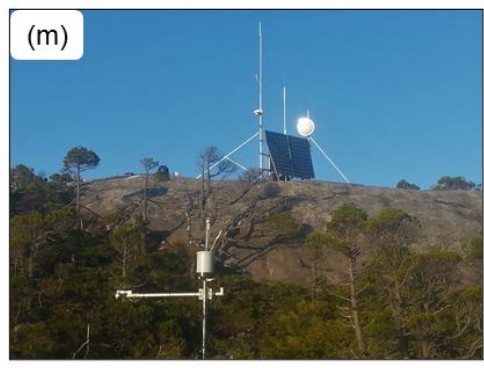

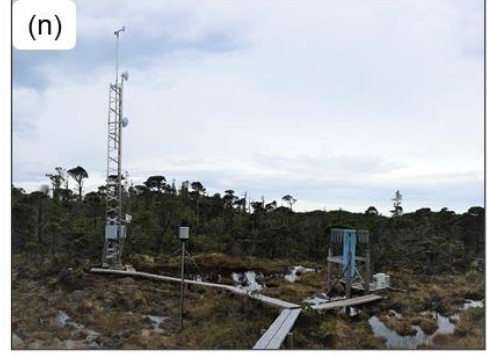

**Meteorological station**

a) SSN626
b) WSN626
c) SSN693
d) WSN693_703
e) WSN703
f) WSN703_708
g) SSN708
h) SSN819
i) SSN1015
j) WSN844
k) WSN819_1015
l) East Buxton
m) Hecate
n) Reference Station






**Figure A2: Images of the seven streams near the watershed outlets. Shown are the water level gauging site (watershed 693); the downstream sites of the automated salt in solution discharge measurement method (watersheds 703 and 1015); the upstream sites of the automated salt in solution discharge measurement method with dumping mechanism (all other watersheds). Photos by Shawn Hateley, Bill Floyd, and Maartje Korver.**


Watershed

a) 626
b) 708
c) 703
d) 693
e) 1015
f) 819
g) 844



## 10 Author contributions

M.K. wrote the manuscript, with contributions from E.H. for the meteorology sections and overall contributions from W.F and I.G.. W.F. designed the hydrometeorological station network and is responsible for installation and maintenance. M.K. and E.H. managed, quality controlled and conducted analysis on the streamflow and weather data respectively, with oversight from W.F.. Data collection, station installation and maintenance was performed by W.F., M.K. and E.H. among many others specified in the acknowledgments. I.G. led the watershed characterization in addition to co-conceptualizing and co-designing the broader Kwakshua Watersheds Observatory along with W.F. and others (see acknowledgments).

## 11 Competing interests

The authors declare no competing interests.

## 12 Acknowledgements

We gratefully acknowledge that this work was conducted on the traditional, ancestral and unceded territories of the Haíɫzaqv and Wuikinuxv Nations. Funding for the Hakai Institute is provided by the Tula Foundation. In-kind and financial support was also provided by the BC Provincial Governments Ministry of Forests, Lands and Natural Resource Operations and Rural Development and Environment and Climate Change Canada for the 'East Buxton' station, which is part of a regional high elevation weather station network. We would like to sincerely thank the Hakai Institute support staff involved in operating the Kwakshua Watersheds Observatory. The installation and development of this observatory arose from research ideas first conceived by Prof. Ken Lertzman, who still acts as a primary scientific advisor. Ray Brunsting was as head of IT a key person in the early days of observatory design and he continues to aid in the management and quality assurance of the data. Hakai Energy Solutions, specifically Colby Owen and James McPhail, have been indispensable in designing, data logger programming and installing the extensive telemetry network enabling real-time data access. Shawn Hateley has led many field maintenance operations and Stewart Butler has been responsible for the maintenance of the salt dilution systems. GIS calculations of watershed characteristics were done by Santiago Gonzalez Arriola. Many people assisted installation and maintenance operations including Will McInnes, Jason Jackson, Rob White, Isabelle Desmarais, David Norwell, Christopher Coxson, Christian Standring, Libby Harmsworth, Ben Millard-Martin, Carolyn Knapper, Midoli Bresch, Parker Christensen, Andrew Sharrock, Darren Cashato, Michel Stitger, Mike Mearns, Dave Snow, Nelson Roberts, Lawren McNab, Ondine Pontier, Eric Courtin and Hannah McSorely.



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
