# Peer review of "High-resolution streamflow and weather data (2013-2019) for seven small coastal watersheds in the northeast Pacific coastal temperate rainforest, Canada"

_Earth System Science Data, 2021_

## Author Response (AR1)

Author comments and responses to peer-review:

Anonymous referee #1

I found the article generally well-written, informative and supporting of the published data set. These data are unique and fill a needed hydrometric gap in the region. I was able to download and read the data and processing scripts. The data appear usable and include minimal metadata. The quantified uncertainty and measurement methods are well-documented and make this data set high-quality. I did find a few aspects confusing and would like to make recommendations for improvement.

Thank you very much for your appreciation of our work and your constructive feedback. Your careful assessment of both the dataset and the article has – in our opinion - much improved our work.

1. Strictly speaking, GitHub and Google Drive are not intended to permanently store scientific data. I would suggest the authors consider using a dedicated data repository to host their data. Data hosting for academic and scientific data can be found through organizations like repositoryfinder, HydroShare, FAIRsharing, and re3data among many others.

We appreciate this comment and agree with the underlying intent to ensure a high-quality data archive. Our data are archived according to the Hakai Institute's data publication standards and norms. The Hakai Institute repository is registered with DataCite Canada and the re3data.org organization mentioned by the reviewer (http://doi.org/10.17616/R31NJMP7). Given the reviewer comment, we reviewed the ESSD policies and concluded that our approach should meet the minimum repository criteria of the journal: https://www.earth-system-science-data.net/policies/repository_criteria.html. We agree that a domain specific repository like HydroShare is a good option yet feel it is not essential to meet the ESSD policy requirements. Finally, we note that the Hakai Institute registered data repository has been successfully used to publish earth system science related papers in several other respected journals including 'Hydrological Processes' and 'Biogeosciences' (i.e. Giesbrecht et al., 2021; St. Pierre et al., 2021).

2. When discussing land use history, I did not see any mention of forest age or specific details about what exactly constitutes "non-vegetated" and "non-forested" land covers.

Thank you. We have added information on forest age based on recent dendroecological work and clarified the land cover types as described below. We provide a general definition of non-vegetated and non-forested land covers illustrated with examples. We refer the reader to the source publication, Thompson et al. (2016), for further details regarding the derivation of these landcover units.

L78: Replaced "Forested, non-forested and non-vegetated landcovers were calculated as % of land in each watershed (total watershed area minus total lake area), using…" by "Forested, non-forested (mainly wetlands with short vegetation) and non-vegetated landcovers (mainly exposed bedrock) were calculated using…"
L80: To further define landcover types, we specified that our wetland class includes forested wetlands. We added the following parenthetical statement: "…wetland cover (including forested wetlands)".
L96: The following sentence was added after the sentence ending with "… forested." "Old-growth forests are extensive and tree ages commonly exceed 250 years (Hoffman et al., 2021)."
L103: Replaced "The watersheds are on average 68 % forested, 24 % non-forested but vegetated (including open wetlands), 4 % non-vegetated (including exposed bedrock), and 4 % covered by lakes (Table 1 )." by " The watersheds are on average 68 % forested, 24 % non-forested but vegetated (mainly wetlands with short vegetation), 4 % non-vegetated (mainly exposed bedrock), and 4 % covered by lakes (Thompson et al., 2016) (Table 1).

3. Soil characteristics are mentioned in passing, but I found no information about soil texture, porosity, or saturated hydraulic conductivity.

Thank you. Unfortunately, we do not have measurements of soil porosity or saturated hydraulic conductivity, but we can qualitatively describe the hydrologically relevant soil characteristics in the following way:

L41: Removed: "soil hydrology,".
L89: Removed: "with shallow (typically < 1 m) but organic-rich soils (Oliver et al., 2017)" and added the following text in line L91: "The soils are thin (<70 cm on average), organic-rich, and have formed in sandy colluvium and patchy morainal deposits (Oliver et al., 2017), resulting in a soil landscape with limited water storage potential."

4. I was confused by the inclusion of the ClimateNA model reference values. I understand this is an observation focused article, not a model validation study. Without corroborating independent observations, I'm not sure the model comparison lends anything to the article.

Thank you for this comment. As reviewer #2 also commented on the length of this paper, and both reviewers commented on superfluous information in the tables, we agree that the model validation is not a necessary analysis to include. ClimateNA data has been used in previous publications of our research group (e.g. Oliver et al., 2017; St. Pierre et al., 2021), as in-situ hydrometeorological data were not available at those times, and we therefore thought it useful to put those modelled data into perspective against our observational data. However, this intention was not clearly communicated and after reconsideration we agree to remove the entire validation analysis of ClimateNA data from this article in the following way:

We removed L353 – L363, as well as column "MAT Clim NA" of Table 2 and "MAT 1981-2010" of Table 4. Note that we did not remove the columns "MAP (PAS) Clim NA" of Table 2 and "MAP (PAS) 1981-2010" of Table 4, as we do not have measurements of "precipitation as snow" for every station.

5. Some information about snow density, texture, or quality may be useful for readers attempting to estimate snow water equivalent from snow depth measurements.

We agree that this would have been useful, but we have been unable to take direct snow measurements due to access restrictions: snow is only persistently present in the winter season above ~700 m elevation. As the study site is a remote island without any transportation infrastructure, these mountain areas are only accessible by helicopter, but winter storm conditions make this trip near impossible. We clarified this in the following way:

L146: Replaced "Snow depth was measured at three high elevation stations (449 – 740 m a.s.l.) (SR50A: Sonic Distance Sensor, Campbell Scientific, Edmonton, Canada) and…" with "Continuous measurements of snow depth were taken at three high elevation stations (449 – 740 m a.s.l.) (SR50A: Sonic Distance Sensor, Campbell Scientific, Edmonton, Canada). Due to access limitations (remote environment and adverse winter weather conditions), snow density observations could not be taken."

6. The automated salt dilution measurement method for high flow discharge was very interesting to read about! I do wonder about potential changes in baseline salinity due to sediment mobilization during high flow events. What advice might the authors have for anyone attempting to duplicate this technique, especially in coastal regions?

Yes during high flow events, baseline salinities could temporarily (over the course of hours) increase due to mobilization of organic material and sediments. As a result, high-flow measurements were often found on either the rising or falling limbs of these natural salinity waves. Therefore, we did not use fixed, but shifting baseline salinities: a linear interpolation of the river's natural salinity at the start and at the end of each salt-dump wave. We did many test runs (i.e. manual salt dumps) to assess the necessary amount of salt (not exceeding environmental thresholds) needed to induce a clear salinity wave at all water levels, easily distinguishable from natural salinity fluctuations.

This approach should work for other coastal river systems, however with a limitation to the size of the stream: measurements should be rapid (< 30 min) to be able to capture a clear salinity signal and to reduce the amount of baseline salinity fluctuations during the measurements. We advise anyone wanting to duplicate this technique to perform ample test dumps before installing equipment, to make sure that salt waves come through clearly.

In addition, we agree that changes in baseline salinity are a source of measurement uncertainty hard to quantify. However, the automated salt dilution system allows for repeated measurements at similar stages, which should capture unaccounted sources of uncertainty by introducing noise into the rating curve. Plotting confidence intervals around that noise as described in this article should then give an indirect quantification of these uncertainties.

We clarified the text in the following way:

*L197: Replaced "…a signal is sent to release a predetermined volume of salt solution. This volume is targeted to never exceed the most sensitive toxicity threshold of 400 mg L-1 (Moore 2004a, 2004b)." by "…a signal is sent to release the salt solution. The volume of solution was predetermined (higher volumes for higher stage levels) to induce a clear signal on top of potential natural salinity fluctuations, but never exceeding the most sensitive toxicity threshold of 400 mg L-1 (Moore 2004a, 2004b)."*

7. My impression from the article is that these were perennial streams. However, I found 0 discharge values in the data set. How should we interpret these 0 values?

We appreciate your careful assessment of our dataset. Although the rivers approach dry conditions in summer, they never go completely dry (stage never reaches 0) and 0 discharge values should therefore not have been included into the timeseries. They were introduced because we extrapolated our rating curves down from the lowest discharge measurement to 0 flow at the lowest measurement of stage. In addition, we used a 3-digit precision for our timeseries (see comment #8), and flows <0.001 were rounded to 0. To resolve these issues, we have updated our dataset in the following ways: 1) we updated the low-flow end of the rating curves by linearly extrapolating from the lowest discharge measurement down to 0 flow at 0 stage and updated the timeseries accordingly. We are aware that assuming 0 flow at 0 stage is also not always valid (e.g. hyporheic flow, pools) but as our rivers never actually reach 0 stage we think that this approach provides us with an acceptable estimation of very low flows. 2) we set the discharge precision at 0.0001 and rounded the entire timeseries to 4 digits.

We recalculated all summary values of discharge mentioned in the article (e.g. annual runoff totals), but as discharge in our rivers are dominated by high-flow events, the changes that had to be made were minimal (i.e. summary values stayed the same or the change was <1% from original value).

*L272: replaced "…estimating minimum stage to equal zero flow…" to "…estimating zero stage to equal zero flow…".*

8. How did the authors decide on the 0.001 m3 s-1 discharge precision? This may have implications on minimum runoff precision for each catchment and the comparisons that can be reasonably made across these catchments, specifically during low flow.

Thank you for noticing this. We agree that a precision of 0.001 is not high enough to accurately present low flows (see also comment #7). We have updated the timeseries by rounding all values to 4 digits.

9. The pressure transducer listed for discharge has an accuracy of +/- 0.05% FS. In the 0-4 m depth range, I think this would correspond to an accuracy of +/- 0.2 cm. However, I did not find any discussion of how this source of uncertainty may (or may not) have influenced discharge uncertainty, especially during low flow.

This is correct. There are two instances where the pressure transducer accuracy can be considered for calculating discharge uncertainty: 1) the uncertainty in the stage timeseries and 2) the uncertainty in the stage value of a stage-discharge measurement on the rating curve. For point 1, a stage uncertainty of +/- 0.2 cm always translated into a discharge uncertainty well within the limits of the rating curve confidence bands and therefore did not need to be added as an additional source of uncertainty. For point 2, we used the standard deviation of stage values recorded during a discharge measurement, rather than the sensor accuracy to calculate stage uncertainty. During low flows/stable flow conditions this generally resulted in an uncertainty very close to 0.2 cm, and at higher flows always >0.2 cm (up to 4 cm in extreme cases). Therefore, incorporating pressure transducer accuracy into our uncertainty analysis would not change final uncertainty estimations. We clarified the text in the following way:

*L274-283: Heavily edited this paragraph, which now reads:*
*"Following the methodology proposed by Coxon et al. (2015), 95 % confidence intervals (CI) were plotted around the rating curves in two steps: first, the absolute error of each stage-discharge measurement was calculated by error propagation of the discharge measurement uncertainty (described above) and the stage measurement uncertainty. Stage measurement uncertainty was calculated as two times the standard deviation of stage values recorded during the discharge measurement, so that measurements taken during rapidly falling or rising water levels got assigned higher uncertainties.* **The accuracy of the stage sensors used (+/- 0.2 cm) was found to be inconsequential to the final calculation of discharge uncertainty, compared to flow turbulence and rapidly changing water levels.** *Second, CIs were derived from 500 curve fitting results of LOESS regressions on 500 randomized sets of stage-discharge measurements and their maximum and minimum absolute measurement errors. For each mm of stage on the rating curve, 500 discharge values and 500 standard deviations were predicted and combined in a Gaussian mixed model, to derive minimum and maximum absolute discharge (95 % CI). Any minimum flow extending below zero was set to zero. "*

Some of the tables and figures seem a bit disjointed (superfluous even). I wonder if Tables 2, 3, and 4 could be heavily edited and resummarized more succintly. The meteorological patterns shown in Figures 3 and 4 are sufficienctly discussed in the text. I'm not sure these figures add anything.

Tables: We agree that the tables contain some superfluous information, that there is some repetition of values across tables, and that generally the purpose of each table is not clearly defined. We suggest to make the following edits:

*More clearly define the purpose of each table in the table captions: "Table 2: Spatial variability of weather variables by meteorological station…", "Table 3: "Annual variability…", "Table 4: Hydrometeorological characteristics of the seven gauged watersheds…".*

*Table 2:*

- *Remove columns "Date installed", "Lat", "Long", and "Elev." Instead, create a new table (A2) dedicated to showing these station specifications.*
- *Remove column "MAT – Clim NA" (see comment #4).*

- *Remove column "MAQ" as these values are repeated in Table 4. This also makes Table 2 dedicated to displaying weather station values only (not stream station), which is cleaner.*

*Table 3:*
- *Remove column "Total discharge" as this is not necessary to include in addition to "total runoff". Also remove "mean streamflow" as this does not change much between years, and watershed average streamflow can be found in Table 4.*
- *Remove columns 'Mean RH' and 'Mean wind speed' as values change very little between years and therefore do not add much information. Added to L349: "Between water years, average precipitation and air temperature varied, whereas wind speed and relative humidity were consistent (2.7 m s-1 and 82 – 84 %) (Table 3).."*

*Table 4:*
- *Remove column "MAT 1981-2010" (see comment #4).*
- *Remove column "MAT" as temperature differences between watersheds are very similar and do not explain differences in runoff processes.*
- *Remove column "MAQ (x10^6 m^3)" as this can easily be calculated from "MAQ (mm)"*
- *Rename "MAQ (mm)" to "MAR (mm)" to clarify that this is runoff, and rename "MAQ (m3 s-1)" to "MASF (m3 s-1)" to clarify that this is streamflow.*

Figures: We agree that figures 3 and 4 do not add any essential information to what is discussed in the text. We will remove both figures and add the following text to clarify how we calculated the lapse rates:

L314: after "by about 120 mm km-1 on Calvert Island" added "(R2 = 0.54, for stations between 'Reference Station' and 'East Buxton')" and after "and 150 mm km-1 on Hecate Island" added "(Fig. 3R2 = 0.88, for stations between 'SSN1015' and 'WSN844')"
L316-317: Added R2 values of lapse rates
L326: Added R2 values of lapse rates

Anonymous referee #2

The pre-print introduces a unique data set from the Pacific coast, Canada. The text is well-written and understandable and the structure of the paper is – in general – OK for me. However, from my perspective the paper is too long and some extensive data analyses could be transferred to a supplement / or appendix. Also, some Tables and figures can be merged showing actually the same information two or more times. I recommend publication after some minor-to-moderate revisions.

Thank you for your appreciation of this dataset. We are very grateful for all the detailed technical comments, and for pointing out the need to be concise in our tables, figures and text. We hope that we have addressed your concerns sufficiently below.

Comments

Title: Daily data? Hourly data? What exactly are weather conditions? Suggestion: Hourly streamflow and weather data (2013-2019) for seven headwaters in Northeast Pacific coastal temperate rainforest, Canada

We will change the title as per your suggestion. Except, instead of 'hourly' we will use 'high-resolution' as we updated our dataset to also include 5-minute data (following your advice in comment #L230). Instead of 'headwaters' we prefer the term 'small coastal watersheds' to emphasize the direct discharge into the coastal ocean. The new title has been updated to:

Replace title with: "*High-resolution streamflow and weather data (2013-2019) for seven small coastal watersheds in the Northeast Pacific coastal temperate rainforest, Canada*".

Structure of the abstract: The two sentences "Measuring rainfall and streamflow […] varied by gauging location" (L19-22) are methods, move them up before giving results in the abstract. Please give the URL to dataset in the abstract. Is the last sentence in the abstract really needed?

We moved sentences L19-22 up, after L15: "…coastal watersheds".
We are happy to adapt the last sentence and include a URL to the dataset in the abstract, however we are unclear whether this would be according to the ESSD guidelines. We would appreciate it if the editor could clarify this for us.

Title suggests "BC in Canada", abstract refers to NPCTR, could you make this consistent?

Yes, thank you. We have removed the reference to BC from the title.

L63-65: Is this method linked to the sentence before? If so, please connect both sentences in a better way. Instead of claiming the new method (again and again) it would be nice to get some insights in the concept of the method (not detailed, but what is the clue here?).

To address this comment, we heavily edited the paragraph starting at L57 as follows, which we hope is more on-point:

*"This article provides a summary of streamflow and weather conditions between 1 October 2013 and 30 September 2019 from the seven watersheds of the KWO that are representative of the outer coast of the NPCTR. In addition, this article highlights how automation and the use of novel technologies made it possible to measure weather and streamflow in a remote environment with access limitations, complex topography, intense storms, and rapid streamflow responses, and special attention is given to the uncertainties associated with these conditions and measurement methods."*

L49: How can those rather short data series be used in the context of climate change?

In L50 we explain why we think that the KWO is particularly well-suited to monitor hydrometeorological responses to climate change with "*as the monitoring program is set up to be long-term and the gauged watersheds are relatively undisturbed*". So the KWO is set-up to continue monitoring and extending this dataset for the long-term: in time, an updated version of this dataset will be very useful, as long-term records of hydrological systems experiencing minimal human disturbances are very sparse (as described by Whitfield et al., 2012). In addition, although we agree that these time-series as published now are too short for analyzing changes over time, they can still be used to establish current baseline conditions. In fact, the data are currently being used to update the PRISM regional climate model (PRISM Climate Group).
To clarify this we made the following changes to the text:

*L48: After "(Dery et al., 2009)" modified the text to: "The KWO monitoring program is set up to be long-term, so the hydrometeorological dataset here presented will be regularly updated to make future analyses of these anticipated system changes possible. The relatively undisturbed environment of the gauged watersheds makes the KWO particularly well-suited to accommodate climate change research programs and to potentially serve within national reference hydrologic networks (Whitfield et al., 2012).*

1: Inset should be highlighted with a rectangle. I guess Fig.1 and Table 1 should be moved in Chapter 2.

We appreciate the importance of highlighting the location of Calvert Island on the regional map. In this case, a rectangle drawn to scale would be very small and hard to see. Instead, we offer a larger (not to

scale) symbol to represent the point location of the observatory in the context of the NPCTR region we refer to in text.

We refer to Fig. 1 in the introduction (Chapter 1) and therefore think the placement of the Figure is correct. However, we agree that Table 1 is better suited in Chapter 2 and will move it.

L102-123: That information is given in Table 1. I suggest so remove or to shorten this section or to focus here more on the differences or specific features of the catchments.

Thank you. We hope that we have addressed this comment by rewriting this paragraph in a more concise way:

*Removed L105-L123 and added the following after L105 "...by lakes (Table 1)." : "Contrasting watershed features are summarized by 1) the topography, where 626, 708, and 1015 drain overall low gradient terrain, 819 and 844 reach medium elevations (465 and 495 m a.s.l.), and 693 and 703, the largest watersheds, drain Mount Buxton (1012 m a.s.l.), which is the only area experiencing permanent seasonal snow cover; 2) the presence of lakes, where 1015 and 708 encompass a large lake (28 - 30 ha), located centrally and near the outlet respectively, watershed 693 is characterized by a chain of four larger lakes (> 5 ha) near the outlet, numerous small lakes and ponds (< 1 ha) are scattered across 626, 708, and 703, and watersheds 819 and 844 have almost no lake cover; and 3) the landcover, where the Hecate Island watersheds are overall more forested than the watersheds on Calvert Island (more wetlands with short vegetation), 626 stands out for its relatively sparse forest cover and high amount of wetlands/unvegetated areas, and watershed 703 is largely unvegetated at high elevations (exposed bedrock) (Table 1)."*

L159: One station is twice as high as the other stations – has this effect be analyzed?

One station has indeed been installed at 4.5/4 meters above ground ('East Buxton') vs. the 2 meter standard, as this station receives seasonal snow and we wanted to avoid the sensors being buried. We have not analyzed the potential effects of this height difference, however for precipitation it has been accounted for with the wind-induced undercatch calculations and corrections of the data (see L231-L235). Further on, we believe that by providing the heights of all stations, we have enabled the data user to analyze or correct for any potential effects themselves, i.e. most models have options to adjust for station above-ground heights. We hope that this clarifies and satisfies the question.

Table A2: ROC and constant value columns are in hours? Column names should be more precise here.

Thank you, we have updated the Table A2 headers and caption as follows:

*Caption: "Thresholds used to instigate quality control procedures of the weather data: maximum and minimum values, a maximum rate of change (δ), and the maximum number of timesteps with a constant value. Outlier data were flagged or where necessary, corrected and gap filled."*
*Headers: "Variable, Unit, Max. (unit), Min. (unit), δ (unit h-1), No. of timesteps with constant value (h)"*

L230: Data is collected on high temporal resolution. Why is now 5, 10 or 15-minutes series available? Wouldn't such a high-resolution data set be interesting considering the rather short length of the time series (i.e. few years).

Thank you for this recommendation. We had originally decided against providing 5-min data (the time-step all our data is generated as) because the rain data was only corrected for wind-induced undercatch at the hourly timestep (the wind data is too noisy at the 5-min level). For consistency across the dataset we therefore chose for hourly timesteps only. However, after reconsideration we agree that there is

much value in providing 5-min data as well, so we have updated our dataset accordingly. We do want to caution the data user however, that there is a discrepancy between the 5-min (uncorrected) and hourly (corrected) precipitation datasets, which we have addressed as follows:

*L66: added "...five-minute and…" before "...hourly timesteps".*
*L234: after "...Legates et al. (2004)." added: "This correction was applied to the hourly timestep only, as wind data at the 5-min level introduced too much noise to the corrected dataset."*
*Dataset: added disclaimer about uncorrected 5-min rain data vs corrected hourly rain data in the 'data_overview.pdf' document.*

If streamflow rating curves are updated every year how is ensured that the derived streamflow flux is comparable across the years? (L243, L494ff)

Thank you for this question. 'Updating a rating curve' can mean two things: 1) new measurements plot nicely on the existing curve (within the 95% confidence bands), which means that no rating curve shift (i.e. streambed morphology changes) has happened. If streambeds are stable (as is for example the case in our watershed 626), the same rating curve can keep on being build or 'strengthened', and is applied to the entire timeseries across years. 2) new measurements plot outside the existing curve, indicating a rating curve shift. In this instance, it is assessed which large storm was responsible for changing the streambed morphology, and a new rating curve is applied to the timeseries starting from the date of this storm. So in both instances, updating a rating curve once a year does not mean that the timeseries are calculated differently every year. Rather, the fact that we update our rating curves so regularly, reduces the chance that a rating curve shift went unnoticed and therefore introduces error into the timeseries. We clarified this in the following way:

*L241: Removed: "Stage-discharge rating curves… 5 to 50 new measurements."*
*L267: After: "…visually assessed and selected" added: "Between 5 to 50 new measurements were added to each station's rating curve every year. Measurements prior to and after high-flow events were analyzed for possible shifts in the rating curves; in case of a shift, time-lapse videos and photos taken during maintenance surveys were investigated to confirm a concurrent change in streambed morphology, and a new curve was applied to the discharge time-series from the onset of the high-flow event."*

L245: Expert correction or automatic routine to correct data?

Stage data were always visually inspected and manually corrected by expert opinion, except for data gaps <30 min, which were automatically gap-filled by linear interpolation. We clarified this in the following way:

*L244: after "…where necessary…" added: "…manually flagged or corrected. As an example, time-lapse photos were used to confirm debris blockages to explain and correct unexpected small fluctuations in stage." And removed "Time-lapse photos were also used to assess the turbulence of flows affecting sensor measurements."*
*L245: Replaced: "Data gaps < 30 min were filled using linear interpolation while longer gaps were filled using linear regression" by: "Data gaps < 30 min were filled using linear interpolation (computer automated) while longer gaps were filled manually using linear regression…"*

L 292: Why is the series now 2015 to 2019? Is 3246 mm the average annual precipitation across all catchments? Why not mm/a?

Thank you, yes we are referring to annual precipitation. The caption and footnotes of Table 3 explain how annual averages were calculated. For the meteorological stations specifically: "Rain, air temperature and relative humidity data were averaged over all meteorological stations, except 'Reference Station' (no data 2015-2016), and 2014-2015 was not calculated because stations 'WSN6226' and 'WSN844' had not been installed yet." And in the footnote "Averaged over water years 2015-2016 to 2018-2019". We have clarified the text in L291 as follows:

*Replaced: "Precipitation from 2015 to 2019 averaged 3246 mm over all watersheds," by: "Annual precipitation was 3246 mm (average of all stations within watershed boundaries, Table 3),…"*

3: The figure suggests a relationship between station precipitation and distance to reference station. Although there is a relationship, I do not understand the purpose of this figure. I guess the purpose is to show the spatial difference in precipitation. Is the relationship even stronger if you replace distance with other attributes of the stations and the reference station (e.g., elevation)? So, Fig 3. and 4 can be merged.

Yes, we agree that these figures were not a necessary addition to the text, and we have therefore decided to remove both figures entirely. Please see the last comment of reviewer #1 where we indicate how we adjusted the text to clarify how we calculated these lapse rates.

5 It might be hard for people with color-vision deficiencies (CVD) to distinguish the 6-7 colours in this Fig. There is a typo in y-label for Air temperature. Perhaps you can arrange these 4 panels in one row and use a wide additional Figure in a second row to show Q and P over the six years (monthly or daily or cumulative for each year). Or even Fig. 7. is enough?

Thank you for noticing the typo, we will correct. We also appreciate your consideration of possible difficulties for people with CVD: we have tested the color scales used in this figure using the Coblis – Color blindness simulator (https://www.color-blindness.com/coblis-color-blindness-simulator/). Still, we acknowledge that distinguishing every single data point on the graph might be difficult. However, we would like to keep the graph the way it is for the following reasons:

First of all, the purpose of this graph is to show average seasonal patterns, as well as interannual variation. The most important features to be distinguished are the average seasonal pattern (thick black line), and the years with large monthly outliers (colored lines). Although maybe not every single data point can be accurately read, we think that these features can be distinguished without much problem. We have tried different visualizations (e.g. the use of point symbols, different colors) but these did not work out any better.
Further on, we think that Figures 5 and 7 show very different aspects of the dataset which would not be captured by Figure 7 only: Figure 5 shows seasonality, whereas Figure 7 illustrates catchment responses of one large storm event. We consider Figure 5 a key figure and arranging its panels in one row would not allow us to recognize seasonal patterns or interannual variation.

On a related note, we have clarified that seasonal patterns were displayed for one station only (and not station averaged) in the text by:

*L335: Added "Seasonal variations in weather variables were analyzed for station 'SSN708', the station with the longest record (installed September 2013, 12 m a.s.l ):"*

6.: Better use a continuous color map and break this into bins. The classes Red and green (also hard to distinguish with CVD) are not there or? Might be helpful to have bins with equal N.

Thank you and we agree. We will update this figure to display bins with approximately equal N and a CVD friendly color scale.

Table 3: For me it is not 100% clear what is meant by runoff, discharge and streamflow (here and in other parts of the paper). Please clarify the use and meaning of these variables. What is the difference between total runoff and total streamflow in the data?

Thank you for this comment. We consider 'discharge' to be a *volume* of river water (m3), 'streamflow' to be a *rate* of river flow (m3 s-1), and runoff to be the *volume* of river water ('discharge') *divided by its drainage area* (mm). 'Total discharge' refers to the total volume of water discharging to the ocean in a certain time period (e.g. 'annual total discharge'). The difference between total runoff and total discharge is that runoff is area weighted (expressed in mm) and discharge is a volume.
We specified this in the text in the following way:

*L64: Before: "All data…" added: "It should be noted that throughout this article, 'streamflow' refers to a flow rate (m3 s-1), 'discharge' describes a volume of river water (m3), and 'runoff' is this volume divided by drainage area (mm)."*

8. could be removed or moved to appendix.

Agreed to move this figure to the appendix. We have renamed it 'A3'

Is the data set useable for other comparable landscapes or is the specific situation on the island(s) too specific to extrapolate potential findings to other regions? The added value of the specific location to measure such a data set should be emphasized more.

Thank you for this comment. We believe that this dataset is especially relevant to the region of the NPCTR and that the data are usable to be extrapolated to the thousands of small, ungauged watersheds that characterize the outer coastline. We believe that the introduction clearly frames our dataset within this regional context, e.g. with Figure 1, and especially L28-L40: *"The outer coast of the… outer coast of British Columbia (Fig. 1). "* We have included a reference to a recent study working on this type of regional scaling for more clarification:

*L39: removed "…on the outer coast of British Columbia (Fig. 1)" and added: "…representing an area of >16,000 km2 in the NPCTR with broadly similar climate, topography, and streamflow regimes (Giesbrecht et al., 2022)."*

*Giesbrecht, I. J. W., Tank, S. E., Frazer, G. W., Hood, E., Gonzalez Arriola, S. G., Butman, D. E., D'Amore, D.V., Hutchinson, D., Bidlack, A., Lertzman.K.P.: Watershed classification predicts streamflow regime and organic carbon dynamics in the Northeast Pacific Coastal Temperate Rainforest. Global Biogeochemical Cycles, 36, e2021GB007047, doi:10.1029/2021GB007047, 2022.*

I recommend to transfer some parts of the draft to a supplement (if possible) to reduce the length of the paper. Also, some detailed (and very interesting) discussion on uncertainty and variation across years or locations could be transferred to such a document. From my point of view the data paper is too long and has to be narrowed down to be more readable.

We appreciate this comment, and we have addressed the 'lengthiness' of the paper through earlier comments by removing 1) superfluous information in tables and figures (or moving them to the appendix), 2) removing entire figures (Fig. 3 and 4), 3) removing an entire analysis (ClimateNA validation), and 4) rewriting sections in a more concise matter (site description). In addition, we have made textual edits (more concise) to some sections that contained wordiness, especially section 4.2.

If a reader still finds this article to be on the long side, we believe that the structure and clear headers of sections and subsections allows this reader to selectively read the sections of interest independent from other sections (e.g. the uncertainty analysis is contained in one section (5), and discharge and weather data are always described in separate sections in case a data user is only interested in one or the other). We would prefer to contain all information provided in this article to one document, as analyses provided in a separate supplementary document often get overlooked.

Technical comments:

L62: rephrase "heavy" and/or make it more precises.

Replaced 'heavy' by 'intense'.

L63: "are near impossible" – why?

These words were removed from the introduction following the reviewer's comment 'L63-L65'.
A detailed explanation of why manual measurements of high flows are near-impossible is given in L188 – L191: *"Manually measuring flows at moderate to high flows was a challenge for multiple reasons: rapid streamflow responses to rain events (generally under 12 hours, Table 4); late fall and winter storm occurrences when field crews were only on site periodically; and safety issues with both accessing the hydrometric stations and taking manual stream flow measurements at high water levels."*

L 140: what is combination solar and micro-hydro?

*L139: Replaced "combination of solar and micro-hydro" by: "…combination of solar panels and micro-hydro power systems"*

Table 2: Please clarify, SS streamflow station, WS weather station? Might be better to sort along those two categories.

Yes 'SSN' stands for 'Stream Sensor Node' and 'WSN' stands for 'Weather Sensor Node'. We clarified this in the caption of Table 2 as well as Table A2:

*₀Station names refer to 'Stream Sensor Node' (SSN) or 'Weather Sensor Node' (WSN), followed by the watershed ID, and are ordered by location (west to east on Calvert, then Hecate Islands). 'RefStn' and 'East Buxton' are located outside watershed boundaries and 'Hecate' is located at the 819/844 watershed boundary (Fig. 1)."*

L226: ROC can be confused, may be using 'delta'?

Agreed we changed 'ROC' to 'δ'.

L295: intense windstorms and convert to km/h?

As our dataset contains wind speeds in m s-1, we consistently stay with this unit throughout the article.

*L295: Replaced 'high' by 'intense'*

Table 3: 3h moving averages

*L377: Replaced 'rolling' by 'moving'.*

---

## Author Response (AR2)

Author comments and responses to second round of peer-review.

**Anonymous referee #1**

No comments or suggestions for revisions.

**Anonymous referee #2**

1. Title is adjusted, much better now.

2. Data URL in the abstract is still valuable from my point of view, but editor has to decided.

We have added the doi of the dataset and the doi of the accompanying watershed delineations to the abstract in the following manner:

L23: Replaced "Links to the complete dataset, watershed delineations with metrics, and calculation scripts can be found in Sect. 6 and 7." with "The complete hourly and five-minute interval datasets can be accessed at https://doi.org/10.21966/J99C-9C14 (Korver et al., 2021) and accompanying watershed delineations with metrics can be found at https://doi.org/10.21966/1.15311 (Gonzalez Arriola et al., 2015)."

3. statement about climate change analysis with this (rather short) dataset is now much clearer

4. L195-196: I can understand why the sensors at one station are around 2 meters higher compared to the other stations. However, this must be mentioned very clearly in the manuscript and the meta data as long the effect is not evaluated in the paper.

We agree and we have further specified this in the text by adding the following sentence:

L238: "This adjustment includes a station height variable, thus accounting for potential effects of the elevated total precipitation gauges at 'East Buxton' and 'Reference Station' and rain gauge orifice at 'East Buxton' (i.e. 4 m instead of 2 m)."

This is in addition to mentioning the station height differences in L157-L159 and in the metadata README document provided with the data package.

5. data update with high-resolution data (5min) is appreciated

6. L289: I cannot see the difference between computer-automated linear regression gap filling and manual linear regression gap filling. Please state clearly how long those gaps are.

Note that the computer automated gap-filling was only applied to gaps < 30 min and that linear interpolation (not regression) was used; gaps > 30 min used back-up sensor data (installed in proximity of the pressure transducer) that were converted to pressure transducer location-specific water levels using a relation established by linear regression. We clarified the text as follows:

L249: replaced "while longer gaps were filled manually using linear regression of the data from the backup stand-alone water level sensors that were installed in proximity of the main stage sensors." with "while gaps > 30 min were filled using a relation (linear regression) between data from back-up standalone water level sensors and pressure transducers"

7. minor comments have been addressed appropriately.